# How vulnerable is my policy? Adversarial attacks on modern behavior cloning policies

## Abstract

Learning from Demonstration (LfD) algorithms have shown promising results in robotic manipulation tasks, but their vulnerability to adversarial attacks remains underexplored. This paper presents a comprehensive study of adversarial attacks on both classic and recently proposed algorithms, including Behavior Cloning (BC), LSTM-GMM, Implicit Behavior Cloning (IBC), Diffusion Policy (DP), and VQ-Behavior Transformer (VQ-BET). We study the vulnerability of these methods to untargeted, targeted and universal adversarial perturbations. While explicit policies, such as BC, LSTM-GMM and VQ-BET can be attacked in the same manner as standard computer vision models, we find that attacks for implicit and denoising policy models are nuanced and require developing novel attack methods. Our experiments on several simulated robotic manipulation tasks reveal that most of the current methods are highly vulnerable to adversarial perturbations. We also investigate the transferability of attacks across algorithms, architectures, and tasks and provide insights into the generalizability of adversarial perturbations in LfD. We find that the success rate of the transfer attacks is highly dependent on the task, raising necessity for more fine-grained metrics that capture both the task difficulties and baseline performance of the algorithms. In summary, our findings highlight the vulnerabilities of modern BC algorithms, paving way for future work in addressing such limitations.

## 1 Introduction

Learning from Demonstration (LfD) has emerged as a powerful paradigm in AI and robotics, enabling agents to acquire complex behaviors from expert demonstrations. These techniques are increasingly deployed in real-world scenarios, such as to enable robots in industrial automation and household robotics. However, these policies pose potential security risks since they can be easily maliciously manipulated by adversaries, causing undesired behaviors or even catastrophic incidents. Motivated by these risks, to the best of our knowledge, we are the first to present a systematic study of the vulnerabilities of modern LfD algorithms.

Adversarial attacks are a widely studied area that aims to develop imperceptible perturbations (Szegedy et al., 2013) to change the output of machine learning models. Early work by Szegedy et al. (2013) and Goodfellow et al. (2014) revealed that adding small, imperceptible perturbations to images could drastically alter the prediction of neural network classifiers. Since then, a significant number of works have explored various attack methods and defense techniques against adversarial attacks (Akhtar & Mian, 2018; Zhang et al., 2020; Chakraborty et al., 2021).

However, the robustness of LfD models to adversarial attacks has been largely overlooked in prior research, particularly in the context of robotic manipulation tasks. While previous works have examined adversarial robustness in reinforcement learning (Mo et al., 2023; Sun et al., 2020; Pattanaik et al., 2017; Lin et al., 2017; Gleave et al., 2019), including whitebox attacks (Huang et al., 2017; Casper et al., 2022) and backdoor detection (Chen et al., 2023), the impact of such perturbations on LfD remains underexplored. Attacks in the LfD domain present unique challenges, including temporal dependencies in sequential decision-making and the multimodal nature of actions in complex environments. This paper aims to investigate the vulnerabilities specific to LfD algorithms under adversarial perturbations, shedding light on their susceptibility and resilience in robotic settings.

Additionally, the attacks developed in this study can be utilized for faster reliability estimation (Tit & Furon, 2024).

Jia et al. (2022) showed that adversarial patches can mislead a robotic arm's object detector, causing it to behave in an undesirable manner. Unlike our work examining the vulnerabilities of modern behavior cloning, they targeted traditional object detection models in industrial robots. To the best of our knowledge, the only prior work closely related to this paper is by Chen et al. (2024) who explore adversarial attacks on diffusion policies (Chi et al., 2023). Their method involves attacking the entire denoising process in the diffusion policy, which is computationally expensive. By contrast, we demonstrate effective attacks by manipulating only a few steps of the denoising process, significantly reducing the attack cost (interms of time and compute). In addition, while the Chen et al. only focus on developing attacks for a single type of policy learning algorithm, our research examines the vulnerabilities of several different LfD algorithms and also explores how adversarial perturbations transfer across different algorithms, tasks, and architectures and visual backbones, offering a broader and more comprehensive perspective on the vulnerability of modern behavior cloning and the generalizability of such attacks.

Our work focuses specifically on post-deployment white-box attacks, where an adversary has access to the trained model parameters but cannot modify the training process. This threat model is particularly relevant for open-source robotics systems where model weights are publicly available, a common practice in modern robotics research and deployment. Unlike training-time attacks that aim to corrupt the learning process, our attacks target the inference phase, attempting to cause task failures through carefully crafted perturbations to visual observations. While this may seem like an overly strong adversarial capability, the increasing trend toward open-source release of robot learning systems makes this a practical concern that needs to be addressed.

In this paper, we evaluate the adversarial robustness of several leading imitation learning frameworks, including Vanilla Behavior Cloning (BC), LSTM-GMM (Mandlekar et al., 2021), Implicit Behavior Cloning (IBC) (Florence et al., 2021), Diffusion Policy (Chi et al., 2023), and VectorQuantizied-Behavior Transformer (VQ-BET) (Lee et al., 2024). Among these, IBC and Diffusion Policy have unique design pipelines that cause naive adversarial attacks to largely fail. IBC employs energy-based models to learn implicit policies, offering greater flexibility in learning complex, multimodal behaviors compared to traditional methods. However, because the correct action is selected based on energy distribution rather than a single output during inference, naive attacks struggle to target the correct action. To address this, we introduce a sampling-based attack method that approximates the local energy surface, increasing the likelihood of selecting the desired target action. Diffusion Policy, on the other hand, uses a generative model approach with denoising diffusion techniques to iteratively refine actions, allowing it to capture diverse and continuous action distributions. While existing attacks (Chen et al., 2024) can degrade performance, they require manipulating the entire denoising process, leading to high attack costs. One of our insights is that attacks are most effective at later stages of the denoising process. By applying attacks specifically only on later stages we can significantly improve attack efficiency. Overall, we hope our results serve as an impetus for enhancing awareness of the security and reliability concerns regarding policies learned via behavior cloning and will inspire researchers to develop more robust LfD algorithms.

The primary contributions of our work are as follows: (1) We conduct the first comprehensive study of white-box adversarial attacks on Learning from Demonstration (LfD) algorithms, encompassing both online (PGD) and offline (Universal Adversarial Perturbation) attacks. We evaluate these attacks in both targeted and untargeted settings and highlight the vulnerability of all the LfD algorithms studied in this paper. (2) We propose novel attack formulations for implicit models such as IBC and Diffusion Policy. Our work addresses the unique challenges posed by the iterative action selection process, representing one of the first successful attacks on these implicit policy models. (3) We provide insights into the non-transferability of attacks across LfD algorithms with similar visual backbones, a unique finding that contrasts with trends in computer vision and highlights the distinct nature of vulnerabilities in LfD systems. (4) We find that, out of all the policies we test, Diffusion Policies are the most robust. While recent work (Carlini et al., 2023) has shown that combining a pretrained denoising diffusion probabilistic model and a standard high-accuracy classifier can yield robustness for image classifiers, we are the first to study and showcase the relative robustness of diffusion policies. We provide evidence that this robustness stems from its multi-step prediction process rather than inherent resilience. Our results show that reducing the prediction horizon significantly decreases the adversarial robustness of diffusion policies.

## 2 BEHAVIOR CLONING ALGORITHMS

In this section, we provide background on the Behavior Cloning (BC) algorithms we study in this paper. To provide clarity throughout the discussion, we first define some key notations used across these algorithms. Let $\xi \in \Xi$ represent a set of expert trajectory demonstrations, where $\xi$ is a trajectory consisting of a sequence of state-action pairs $(s, a)$, sampled from an expert policy $\pi^*(s)$. Our objective in behavior cloning is to learn a policy $\pi_\theta(s)$, parameterized by $\theta$, that imitates the expert's behavior by minimizing a loss function $L$ that measures the difference between the expert actions and the actions predicted bu the learned policy.

Formally, for a given policy $\pi_\theta$, we aim to minimize: $\pi_\theta^* = \arg\min_{\pi_\theta} \sum_{\xi \in \Xi} \sum_{s \in \xi} L(\pi_\theta(s), \pi^*(s))$

where $L$ is typically the cross-entropy loss for discrete actions or mean squared error for continuous actions. However, some methods leverage specialized losses: LSTM-GMM uses a log-likelihood objective to capture multimodal action distributions, while VQ-BET incorporates additional quantization losses. In Implicit Behavior Cloning (IBC), the loss is framed as a contrastive energy-based model, and Diffusion Policy relies on a loss function based on noise estimation in a denoising process. Despite these differences, the overarching goal remains the same: learning a policy that best imitates the expert's demonstrations.

### 2.1 VANILLA BEHAVIOR CLONING

Vanilla Behavior Cloning (Vanilla BC) learns a policy via supervised learning (Bain & Sammut, 1995; Torabi et al., 2018). Given a dataset of state-action pairs $(s, a)$, it directly maps states to actions using a neural network trained to minimize the cross-entropy loss for discrete actions or mean squared error (MSE) for continuous actions. While effective for simple tasks, Vanilla BC struggles with tasks requiring long-term dependencies owing to the problem of compounding error and the tasks with multimodalilty in expert behavior, as it assumes a unimodal distribution over actions (Ross et al., 2011; Florence et al., 2021).

### 2.2 LSTM-GMM

Long Short-Term Memory with Gaussian Mixture Model (LSTM-GMM) (Mandlekar et al., 2021) enhances Vanilla BC by incorporating temporal dependencies through an LSTM network (Hochreiter & Schmidhuber, 1997). The LSTM processes a sequence of states $s_1, s_2, \ldots, s_T$ recursively, maintaining an internal hidden state $h_t$ at each time step. The policy $\pi_\theta(a_t|s_t, h_{t-1})$ is parameterized by the LSTM to model the temporal structure, while a GMM captures multimodal action distributions at each time step. At each time step $t$, the LSTM updates its hidden state and predicts a multimodal distribution over actions : $h_t = \text{LSTM}(s_t, h_{t-1})$ and $p(a_t|s_t, h_{t-1}, \theta) = \text{GMM}(h_t)$. The policy is trained by maximizing the likelihood of the observed actions given the state sequence: $\pi_\theta = \arg\max_\theta \sum_{\xi \in \Xi} \sum_{t=1}^{T} \log p(a_t|s_t, h_{t-1}, \theta)$, where $p(a_t|s_t, h_{t-1}, \theta)$ is the probability of action $a_t$ under the GMM, conditioned on the current state $s_t$ and the previous hidden state $h_{t-1}$.

### 2.3 IMPLICIT BEHAVIOR CLONING

Implicit Behavior Cloning (IBC) (Florence et al., 2021) reformulates the problem of policy learning as an energy-based model (EBM). Instead of explicitly predicting actions, IBC defines a compatibility score between states and actions using an energy function $E_\theta(s, a)$. The policy is implicitly represented by selecting actions that minimize the energy: $\pi_\theta(s) = \arg\min_a E_\theta(s, a)$ The model is trained using contrastive learning, where the energy of expert actions is minimized relative to negative (non-expert) samples. The training loss typically follows the InfoNCE objective, as discussed in more detail in section 3.3.1.

### 2.4 DIFFUSION POLICY

Diffusion Policy (DP) (Chi et al., 2023) uses a novel generative approach to model action distributions by leveraging Denoising Diffusion Probabilistic Models (Ho et al., 2020). The policy is represented as a reverse diffusion process, which iteratively refines actions from Gaussian noise towards

the true distribution. Given a noisy action $a_T$ sampled from a Gaussian distribution, the model iteratively denoises it using a learned denoising function conditioned on the state. The policy, with $a_0$ being the final action obtained after denoising, is defined as: $\pi_\theta(a_0|s) = p_\theta(a_T|s) \prod_{t=1}^{T} p_\theta(a_{t-1}|a_t, s)$

## 2.5 VQ-BET

The Vector Quantized Behavior Transformer (VQ-BET) (Lee et al., 2024) combines a transformer-based architecture with vector quantization to handle multi-modal continuous action spaces. The policy discretizes actions into latent codes using a hierarchical quantization process, which allows the model to capture both coarse- and fine-grained action details. The model's policy is formulated as a sequence prediction problem, where the transformer predicts discrete latent codes and continuous offsets for actions.

## 3 ADVERSARIAL ATTACKS ON IMITATION LEARNING

We study two widely used adversarial attacks in our paper, namely, Projected Gradient Descent (PGD) (Madry et al., 2017) and Universal Adversarial Perturbations (UAP) (Moosavi-Dezfooli et al., 2016). PGD iteratively applies a projected Fast Gradient Sign Method (FGSM) attack (Goodfellow et al., 2014). UAP generates adversarial perturbations by considering multiple samples. Detailed explanations about these attacks are given in Appendix A.

### 3.1 THREAT MODEL

Before describing our attack methods, we first clearly specify our threat model:

**Adversary's Goal**: The attacker aims to cause task failure by perturbing visual observations during deployment, either through untargeted perturbations that disrupt normal policy execution or targeted perturbations that force specific undesired actions.

**Adversary's Knowledge and Capabilities**: The attacker has white-box access to the trained policy parameters but cannot modify them. Perturbations are limited to the visual observation space (no direct action manipulation). Perturbations must remain within an $L_p$ norm ball of radius $\epsilon$ to maintain imperceptibility. The attacker can compute gradients through the entire policy network.

This threat model is particularly relevant for deployed robotic systems using open-source policies, where model weights are publicly available but the training process is complete. We study both online attacks (PGD) that can adapt perturbations in real-time and offline attacks (UAP) that must generate a single fixed perturbation designed to work across all states.

### 3.2 ATTACKS ON EXPLICIT BC ALGORITHMS

The objectives for running PGD, and UAP on explicit behavior cloning methods such as Vanilla BC, LSTM-GMM, and VQ-BET are based directly on their respective loss functions. For Vanilla BC, the adversarial attacks aim to maximize the mean squared error (MSE) loss between the predicted and expert actions by introducing small perturbations to the input states. In LSTM-GMM, the attacks target the temporal dependencies modeled by the LSTM and the multimodal action distributions captured by the Gaussian Mixture Model (GMM), aiming to disrupt the likelihood maximization over the GMM outputs. For VQ-BET, the attacks exploit the latent action space by targeting the prediction loss of discrete latent codes, ultimately leading to suboptimal action predictions. Each attack (PGD, UAP) thus aims to create adversarial perturbations that exploit the specific vulnerabilities of these loss functions to degrade performance.

### 3.3 ATTACKS ON IMPLICIT BC ALGORITHMS

Implicit BC algorithms differ from explicit ones in modeling the learning process and action selection, causing naive adversarial attacks largely fail to generate feasible perturbations. In this section, we propose new attack formulations for implicit BC models considering their unique designs.

### 3.3.1 IMPLICIT BEHAVIOR CLONING

Implicit Behavior Cloning (IBC) leverages implicit modeling techniques and contrastive learning to learn a policy directly from expert demonstrations. In IBC, the policy $\pi_\theta(\mathbf{a} \mid \mathbf{s})$ is learned by optimizing an energy-based model (EBM) that assigns low energy values to actions demonstrated by the expert and higher energy values to other actions. The energy function $E_\theta(\mathbf{s}, \mathbf{a})$ parameterized by $\theta$ is trained using the InfoNCE loss, for a batch of $N$ actions:

$$\mathcal{L}_{\text{InfoNCE}} = \sum_{i=1}^{N} -\log \left( \frac{e^{-E_\theta(\mathbf{s}_i, \mathbf{a}_i)}}{e^{-E_\theta(\mathbf{s}_i, \mathbf{a}_i)} + \sum_{j=1}^{N_{\text{neg.}}} e^{-E_\theta\left(\mathbf{s}_i, \tilde{\mathbf{a}}_i^j\right)}} \right) \tag{1}$$

where $\tilde{\mathbf{a}}_i^j$ for $j = 1, \ldots N_{\text{neg.}}$ are the negative samples. The parameters $\theta$ are optimized by minimizing $\mathcal{L}_{\text{InfoNCE}}$, encouraging the model to assign lower energy to expert actions compared to negative samples. This approach allows IBC to capture complex and multimodal action distributions, leading to more robust imitation of expert behaviors (Florence et al., 2021). Since IBC uses an implicit model with iterative sampling procedure for selecting actions, we need to develop specific formulations for untargeted and targeted attacks for these kinds of implicit models, specifically for the Derivative-Free Optimizer version of the inference. Algorithm 1 summarizes our attack for IBC.

For the targeted attack, unfortunately there is no clear end-to-end loss function that minimizes the targeted action energy under the clean state-action distribution, as the negative action samples are sampled randomly and thus the probability of selecting the targeted action can be very low. Hence, we formulate the problem as finding the perturbation $\delta$ for a target action $\mathbf{a}'$ such that, $\tilde{p}_\theta(\mathbf{a}'_i|\mathbf{s}_i + \delta) > \tilde{p}_\theta(\mathbf{a}_i|\mathbf{s}_i + \delta)$. To achieve this, *we introduce a sampling-based attack method to approximate the local energy surface*, making it easier to select the target action. Specifically, we randomly sample a small number of negative actions, along with the target action, to estimate the energy surface around the target region. We also consider the original action during the attack. we then iteratively perform gradient ascent to decrease the energy of the target action compared to both the original and negative actions. However, in-order to further increase the probability of the target action being chosen during inference, we repeat this procedure $N_{iter}$ times to decrease the energy of the target action with respect to more actions that could possibly be selected due to their vicinity to the original actions. The details are presented in Algorithm 1.

For the untargeted attack, the objective is to perturb the state s by finding a perturbation $\delta$ that increases the energy (reduces the probability) of the actions that would normally be taken by the trained policy on clean state observations. In this case, we aim to push the model toward selecting less optimal actions by maximizing the energy associated with the learned actions in the perturbed state. To achieve this, we perform gradient ascent on the input pixels to maximize the energy of the correct action ($a_{clean}$ in Algorithm 1). Thus, the selected actions are essentially random actions with respect to the original state-action distribution.

### 3.3.2 DIFFUSION POLICY

Diffusion Policy (DP) (Chi et al., 2023) aims to overcome the necessity of approximating the normalizing constant (the negative samples required in the above IBC method) in an energy based model, by learning the score function of the action-distribution.

In particular, the score function is defined as the gradient of the log-conditional probability distribution of actions, which is usually learnt as a noise-prediction network ($\varepsilon_\theta$) parameterized by $\theta$.

$$\nabla_{\mathbf{a}} \log p(\mathbf{a} \mid \mathbf{s}) = -\nabla_{\mathbf{a}} E_\theta(\mathbf{s}, \mathbf{a}) \approx -\varepsilon_\theta(\mathbf{s}, \mathbf{a}) \tag{2}$$

Starting from $\mathbf{a}^k$ sampled from Gaussian noise, DP iteratively denoises the sample $k$ times to get a desired noise-free sample $\mathbf{a}^0$.

$$\mathbf{a}^{k-1} = \alpha \left( \mathbf{a}^k - \gamma \varepsilon_\theta \left( \mathbf{s}, \mathbf{a}^k, k \right) + \mathcal{N} \left( 0, \sigma^2 I \right) \right) \tag{3}$$

where $\alpha, \gamma, \sigma$ are the hyper-parameters that collectively define the noise schedule. The complete inference is defined in the Appendix. C.

Algorithm 2 shows our online attack method for Diffusion Policy. Both targeted and untargeted attacks use Mean Squared Error (MSE) loss for propogation of gradient. For the targeted attack, we

try to minimize the distance between our predicted action and the target action (line 10) by doing gradient descent. Whereas, in the untargeted attack, we try to maximize the distance between the predicted action and the clean action (line 12) by doing gradient ascent. Running this attack end-to-end during the whole denoising process can be costly, as we need to backpropogate through the entire network for each iteration for a single inference step. However, we can reduce this computation by taking inspiration from prior work on image editing attacks (Salman et al., 2023) and only apply the perturbations during last timesteps (line 7 of Algorithm 2) of the denoising process. This enables us to avoid wasting attacks when the actions are very random (during the initial steps of denoising) and only apply the attack when the data has started to converge towards the mode. This reduces the attack effort while not affecting the quality of adversarial attack.

---

**Algorithm 1** Implicit BC PGD Attack

---

**Require:** Trained energy model $E_\theta(s, a)$, state $s$, observation $o$, number of samples $N_{samples}$, number of iterations $N_{iters}$, decay rate $K$, perturbation bound $\epsilon$, step size $\alpha$
1: Obtain clean action $a_{clean}$ by running IBC on $s$
2: **for** $epoch = 1, 2, \ldots, N_{epochs}$ **do**          ▷ Optimize over multiple samples
3:  Initialize sample set $\mathcal{S} = \{\tilde{a}^i\}_{i=1}^{N_{samples}} \sim \mathcal{U}(a_{min}, a_{max})$
4:  **if** Targeted **then**
5:    Introduce $a_{clean}, a_{target}$ into $\mathcal{S}$
6:  **end if**
7:  **for** $iter = 1, 2, \ldots, N_{iters}$ **do**       ▷ Inner PGD attack iterations
8:    $\{E_i\}_{i=1}^{|\mathcal{S}|} \leftarrow \{E_\theta(s', \tilde{a}^i)\}_{i=1}^{|\mathcal{S}|}$    ▷ Compute energies with perturbation
9:    $\{\tilde{p}_i\}_{i=1}^{|\mathcal{S}|} \leftarrow \left\{ \frac{e^{-E_i}}{\sum_{j=1}^{|\mathcal{S}|} e^{-E_j}} \right\}_{i=1}^{|\mathcal{S}|}$    ▷ Compute softmax probabilities
10:   **if** Targeted **then**
11:    Compute cross-entropy loss with $a_{target}$ as the true label:
12:    Loss $= -\log(\tilde{p}_{\text{target}})$       ▷ $\tilde{p}_{\text{target}}$ is the probability of $a_{target}$
13:   **else**
14:    Compute untargeted loss:
15:    Loss $= -E_\theta(s', a_{clean})$  ▷ Maximize energy of the correct action for untargeted attacks
16:   **end if**
17:   Update $s'$ using PGD step:
18:   $s' = s' + \alpha \cdot (\nabla_{o_t} \text{Loss})_{\mathcal{B}_\epsilon}$     ▷ Projected on the $l_p$ norm ball
19:  **end for**
20: **end for**
21: **return** $s'$

---

---

**Algorithm 2** Diffusion Policy PGD Attack

---

**Require:** Observation horizon $T_0$, Action Horizon $T_a$, Prediction Horizon $T_p$, State sequence $\mathbf{S}_t = \{\mathbf{s}_{t-T_o+1}, \ldots, \mathbf{s}_t\}$, number of denoising iterations $K$
**Ensure:** Action sequence $\mathbf{A}_t = \{\mathbf{a}_t, \ldots, \mathbf{a}_{t+T_p-1}\}$
1: $\mathbf{A}_t^{clean} =$ Diffusion Policy Inference$(\mathbf{S}_t)$
2: $\mathbf{A}_t^{target} = \mathbf{A}_t^{clean} +$ Desired Perturbations      ▷ Only for Targeted Attack
3: Initialize $T_{\text{attack}}, \epsilon, \alpha, \gamma, \sigma, N_{iters}$
4: Initialize $\mathbf{A}_t^{(K)} \sim \mathcal{N}(\mathbf{0}, \mathbf{I})$
5: **for** $k = K, K-1, \ldots, 1$ **do**
6:  $\mathbf{A}_t^{(k-1)} = \alpha(\mathbf{A}_t^{(k)} - \gamma\epsilon_\theta(\mathbf{S}_t, \mathbf{A}_t^{(k)}, k)) + \sigma\mathcal{N}(\mathbf{0}, \mathbf{I})$
7:  **if** $k < T_{\text{attack}}$ **then**      ▷ Attack during the last $K - T_{attack}$ timesteps
8:   **for** $N_{iters}$ **do**            ▷ Inner PGD iterations
9:    **if** Targeted **then**
10:     $Loss = -\text{MSELoss}(\mathbf{A}_t^{k-1}, \mathbf{A}_t^{target(k-1)})$
11:    **else**
12:     $Loss = \text{MSELoss}(\mathbf{A}_t^{k-1}, \mathbf{A}_t^{clean(k-1)})$
13:    **end if**
14:    $\mathbf{S}_t = \mathbf{S}_t + \alpha \cdot \nabla_{\mathbf{O}_t}(Loss)_{\mathcal{B}_\epsilon}$ ▷ Grad. ascent w.r.t current observations and project on $\epsilon$ ball.
15:   **end for**
16:  **end if**
17: **end for**
18: **return** $\mathbf{S}_t$

---

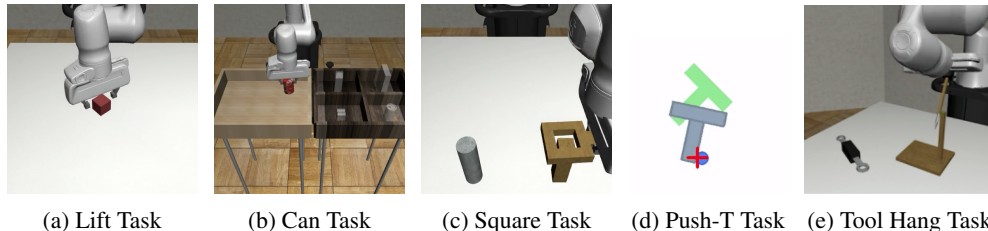

(a) Lift Task     (b) Can Task     (c) Square Task     (d) Push-T Task     (e) Tool Hang Task

Figure 1: Environments used to study adversarial robustness of modern behavior cloning algorithms. (a)-(c) and (e) are from RoboMimic (Mandlekar et al., 2021) and (d) is from Florence et al. (2021)

## 4 EXPERIMENTS & RESULTS

We design our experiments to the answer to following questions: (1) How vulnerable are modern behavior cloning algorithms to adversarial attacks? (2) How easy is it to craft universal perturbations for these algorithms? (3) How transferable are the attacks across different algorithms and different tasks? (4) What is the impact of different feature extraction backbones on attack performance, as in how transferable are the attacks between different vision architectures? (5) How does the action prediction horizon of diffusion policy affect its vulnerability?

### 4.1 ENVIRONMENTS

To demonstrate the adversarial robustness of modern behavior cloning algorithms, we consider common benchmarks shown in Figure 1. The tasks of Lift, Can and Square are taken from Robomimic (Mandlekar et al., 2021), where the state-of-the art frameworks such as Diffusion Policy and LSTM-GMM have been shown to have a nearly 100% success rate in non-adversarial settings. To further assess the ability of adversarial attacks to breach these frameworks on more sophisticated interaction data, we consider the Push-T environment, first introduced by Florence et al. (2021) and then subsequently used by Diffusion Policy (Chi et al., 2023) and VQ-BET (Lee et al., 2024). Descriptions and details of these tasks are included in Appendix B.

### 4.2 PRETRAINED POLICIES

To provide consistent and reproducible restults, we attack the pre-trained checkpoints for LSTM-GMM, IBC and Diffusion Policy released by the authors of Diffusion Policy (Chi et al., 2023) on these suite of tasks, and train our policies for Vanilla-BC and VQ-BET, due to absence of publicly available checkpoints. We evaluate all the environments on 50 randomly initialized environments across 3 different seeds for reporting the mean and standard deviation of the success rate. All pretrained policies and source code for generating and evaluating attacks and reproducing our results will be open-sourced at [*url masked for anonymous submission*].

### 4.3 HOW VULNERABLE ARE MODERN BEHAVIOR CLONING ALGORITHMS TO ADVERSARIAL ATTACKS ?

To assess the vulnerability of modern behavior cloning algorithms to adversarial attacks, we conducted a comprehensive evaluation using both online (PGD) and offline (UAP) attack methods. Our findings, as illustrated in Figures 2 and 3, reveal significant vulnerabilities in the adversarial robustness of current algorithms when faced with perturbations in the observation space. Among the algorithms tested, VQ-BET demonstrated the highest susceptibility to adversarial perturbations. We hypothesize that this vulnerability stems from the discrete nature of its action space, which may lead to discontinuous decision boundaries. In contrast, algorithms employing iterative methods for action selection, such as IBC and Diffusion Policy, exhibited relatively higher robustness. This enhanced resilience can be attributed to the inherent stochasticity in their action selection processes during inference. It is important to note that the effectiveness of these attacks varies depending on the complexity of the task environment. For instance, the Lift environment allows for a larger margin of error, making it more forgiving to substantial perturbations in actions. However, as task complexity increases, we observe a dramatic reduction in the robot task success rates (increase in attack success

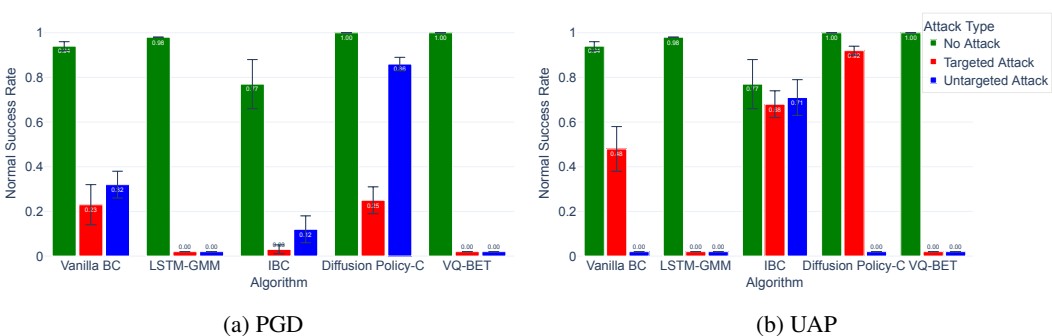

(a) PGD            (b) UAP

Figure 2: Comparison of PGD and UAP attacks for the Lift task. The y axis denotes the normal performance of the evaluated policies, which is the lower the better for attacks.

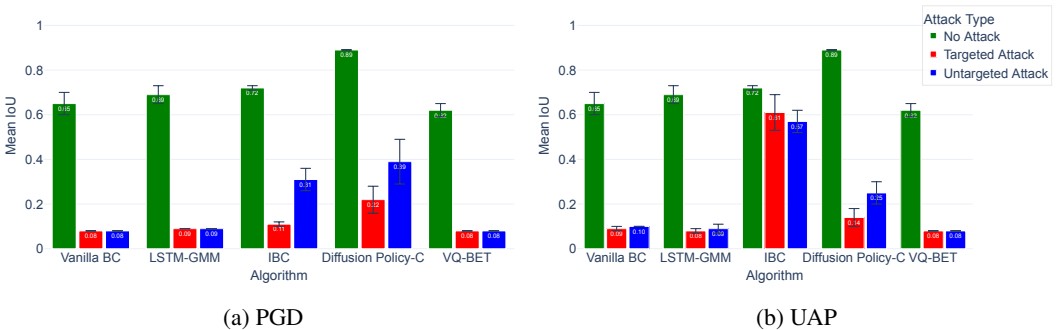

(a) PGD            (b) UAP

Figure 3: Comparison of PGD and UAP attacks for the Push-T task. The y axis denotes the normal performance of the evaluated policies, which is the lower the better for attacks.

rates) across all algorithms. For example, Mandlekar et al. (2021) categorize the difficulty of the tasks with Lift being the easiest, Can being harder than Lift, and Square being harder than Can. As we increase the complexity of the task, we notice an increase in the efficacy of the adversarial attacks as detailed in Appendix E. We also observe that even for small values of epsilon most of the algorithms are not robust to the attacks (Fig. 12 in Appendix J).

## 4.4 CAN ADVERSARIAL EXAMPLES TRANSFER ACROSS DIFFERENT ALGORITHMS AND TASKS?

The transferability of adversarial examples across different behavior cloning algorithms presents an intriguing phenomenon, given the substantial differences in their loss functions and training methodologies (as detailed in Section 2). While these algorithms share a common image encoder (ResNet-18), their end-to-end training approach results in distinct feature representations that are not easily interpretable. The transferability of adversarial examples across different behavior cloning algorithms presents an intriguing phenomenon, given the substantial differences in their loss functions and training methodologies (as detailed in Section 2). While these algorithms share a common image encoder (ResNet-18), their end-to-end training approach results in distinct feature representations that are not easily interpretable.

In simpler environments like the Lift task (see Table 1), where baseline success rates are high (>90% for most algorithms), we observed limited transferability with relatively small proportional drops in performance, aligning with our initial expectations. Intriguingly, as we progressed to more complex environments (Square: see Table 5), where baseline success rates are lower and tasks are naturally less robust to action perturbations, we noticed that transferred attacks often caused larger proportional drops in performance relative to the baseline.

Table 1: Inter-Algorithm Transferability of Untargeted UAP on the Lift task, where the rows correspond to the attacker policy over which perturbations were developed (random refers to a Gaussian noise with the mean of zero and std of epsilon) and the columns correspond to target policy over which attacks were tested.

| Attacker / Target Policy | Vanilla BC | LSTM-GMM | IBC | DiffusionPolicy-C | VQ-BET |
|---|---|---|---|---|---|
| Random | 0.96 | 0.84 | 0.80 | 1.00 | 0.98 |
| Vanilla BC | 0.00 | 0.00 | 0.80 | 1.00 | 0.94 |
| LSTM-GMM | 0.94 | 0.00 | 0.72 | 1.00 | 0.96 |
| IBC | 1.00 | 0.10 | 0.64 | 1.00 | 0.98 |
| DiffusionPolicy-C | 0.82 | 0.22 | 0.78 | 0.00 | 0.94 |
| VQ-BET | 0.94 | 0.50 | 0.84 | 1.00 | 0.00 |

Table 2: Inter-Algorithm Transferability of Untargeted UAP on the Push-T task.

| Attacker / Target Policy | Vanilla BC | LSTM-GMM | IBC | DiffusionPolicy-C | VQ-BET |
|---|---|---|---|---|---|
| Random | 0.60 | 0.61 | 0.64 | 0.82 | 0.55 |
| Vanilla BC | 0.10 | 0.08 | 0.41 | 0.80 | 0.22 |
| LSTM-GMM | 0.15 | 0.09 | 0.33 | 0.78 | 0.31 |
| IBC | 0.14 | 0.08 | 0.14 | 0.71 | 0.17 |
| DiffusionPolicy-C | 0.27 | 0.10 | 0.24 | 0.14 | 0.22 |
| VQ-BET | 0.26 | 0.14 | 0.47 | 0.61 | 0.08 |

Table 3: Inter-Architecture Transferability. Transferring adversarial perturbations generated on ResNet-18 to ResNet-50 as backbone on the Lift task. NA: No Attack

| Algorithm | NA Resnet-18 | NA Resnet-50 | Resnet-18 | Resnet-50 |
|---|---|---|---|---|
| Vanilla BC | 1.00 | 1.00 | 0.21 | 0.75 |
| LSTM-GMM | 1.00 | 1.00 | 0.00 | 0.25 |
| IBC | 0.95 | 0.50 | 0.85 | 0.38 |
| DiffusionPolicy-C | 1.00 | 1.00 | 0.00 | 1.00 |
| VQ-BET | 1.00 | 1.00 | 0.00 | 0.98 |

This analysis highlights the importance of considering relative performance metrics when evaluating transferability across tasks of different complexity. Future work could benefit from developing normalized metrics that better account for task difficulty and baseline performance. Additional experiments and discussion for the inter-task transferability are in Appendix G.

## 4.5 WHAT IS THE IMPACT OF DIFFERENT FEATURE EXTRACTION BACKBONES TO ATTACK PERFORMANCE?

Our investigation into the impact of different vision encoder backbones on adversarial attack transferability reveals intriguing insights. We developed perturbations using ResNet-18 as the backbone and then deployed these attacks on policies than were trained using ResNet-50, without regenerating the attacks. This cross-architecture transfer scenario yielded surprising results. In the Lift task (see Table 3), we observed high transferability for some algorithms (e.g., LSTM-GMM and IBC), while others showed more resilience (e.g., Diffusion Policy-C and VQ-BET). The more complex Push-T task (see Table 4) demonstrated a more consistent pattern of partial transferability across all algorithms. Notably, in many cases, the ResNet-50 models showed vulnerability to attacks developed for ResNet-18, suggesting that simply increasing model capacity does not guarantee improved robustness against cross-architecture attacks. It also highlights the existence of shared vulnerabilities across different network architectures, which adversarial perturbations can exploit even when transferred to a different backbone. These results underscore the importance of considering cross-architecture vulnerabilities in the design of robust behavior cloning systems

Table 4: Inter-Architecture Transferability. Transferring adversarial perturbations generated on ResNet-18 to ResNet-50 as backbone on the Push-T task.

| Algorithm | NA Resnet-18 | NA Resnet-50 | Resnet-18 | Resnet-50 |
|---|---|---|---|---|
| Vanilla BC | 0.72 | 0.62 | 0.09 | 0.20 |
| LSTM-GMM | 0.72 | 0.56 | 0.08 | 0.21 |
| IBC | 0.74 | 0.57 | 0.63 | 0.27 |
| DiffusionPolicy-C | 0.88 | 0.78 | 0.14 | 0.54 |
| VQ-BET | 0.62 | 0.65 | 0.08 | 0.29 |

### 4.6 HOW DOES THE ACTION PREDICTION HORIZON OF DIFFUSION POLICY AFFECT ITS VULNERABILITY?

In addition to the above experiments, we find an interesting trade-off between the action horizon of the Diffusion Policy and robustness. In Fig. 4, we observe that as the action horizon increases (the number of actions taken at a time), while keeping the prediction horizon the same, the policy shows increasing robustness to universal attack. We hypothesize that as the action horizon increases the number of times the perturbed observation gets observed decreases thus allowing for smaller compounding errors during the inference. However, if the action horizon is too long then the latency and recovering from sub-optimal trajectories might lead to worse overall performance.

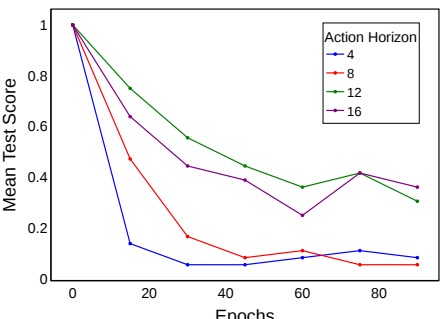

Figure 4: Test mean score vs epochs for action prediction horizon of 16 for lift, and various action horizons during Untargeted UAP.

## 5 CONCLUSION & FUTURE WORK

Our results show that all modern behavior cloning algorithms are vulnerable to adversarial attacks. Interestingly, implicit policies such as Implicit Behavior Cloning and Diffusion Policy seem to be more robust than the explicit policies. However, our results also demonstrate that the attack success rate is dependent on the task. As tasks gets harder it becomes easier to attack these algorithms. This also holds true based on the results from transferability of attacks between different algorithms. Our results provide evidence that the different algorithms and the same algorithm trained with a different architecture are learning some similar features that are not completely orthogonal but also not completely similar. Thus posing a security challenge since even if we are using different vision encoders, task, or policy these perturbations are still transferable.

We believe that our work lays foundation for future work in the direction of adversarial robustness of robotic policies. We also believe that as this field progresses, there is a need for better metrics to capture the nuanced effects of adversarial attacks on trajectories, rather than relying solely on success rates. Such metrics could provide deeper insights into the uncertainty in the state-action distributions learned by the policies. We also think that, while a lot of progress has been made in computer vision interms of developing and patching adversarial attacks, the sequential nature of robotic policies and the non-linearity from vision representations to actions can also be a source of new vulnerabilities. While adversarial defenses such as Randomized Smoothing (Cohen et al., 2019) (see results and discussion in Appendix F) can help in increasing the robustness, it comes at the cost of large increase in the reaction time and may struggle when the action distribution exhibits multi-modality. Additional defenses such as Adversarial Training (Goodfellow et al., 2014) are left for future work.

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

# A ADVERSARIAL ATTACKS

## A.1 FAST GRADIENT SIGN METHOD (FGSM)

Fast Gradient Sign Method was proposed by Goodfellow et al. (2014). The basic idea behind FGSM is to use the linearity of neural networks to craft adversarial examples. It is designed to be fast (on the $L_\infty$ space) instead of a more close or robust adversarial example. Given an image $x$ the method sets the adversarial example as,

$$x' = x + \epsilon \cdot sign(\nabla loss_F(x))$$

Intuitively, it tries to move each pixel by a small amount ($\epsilon$) with the direction determined by the sign of the gradient of the loss function wrt input.

## A.2 PROJECTED GRADIENT DESCENT (PGD)

Projected Gradient Descent (PGD) is an iterative adversarial attack algorithm that generalizes the Fast Gradient Sign Method (FGSM) by applying multiple steps of gradient ascent to maximize the loss function with respect to the input, subject to a constraint on the perturbation magnitude. Mathematically, starting from an initial input $\mathbf{x}_0$, the PGD algorithm iteratively updates the input $\mathbf{x}_{k+1}$ using the following rule:

$$\mathbf{x}_{k+1} = \Pi_{\mathcal{B}_\epsilon(\mathbf{x}_0)} \left( \mathbf{x}_k + \alpha \cdot \text{sign}(\nabla_{\mathbf{x}} J(\theta, \mathbf{x}_k, y)) \right),$$

where $J(\theta, \mathbf{x}_k, y)$ is the loss function of the model with parameters $\theta$, input $\mathbf{x}_k$, and true label $y$; $\alpha$ is the step size; and $\Pi_{\mathcal{B}_\epsilon(\mathbf{x}_0)}$ denotes the projection operator onto the $l_p$-norm ball $\mathcal{B}_\epsilon(\mathbf{x}_0)$ of radius $\epsilon$ centered at $\mathbf{x}_0$. The projection step ensures that the perturbed input remains within the allowable perturbation bound. When the number of iterations is set to one and the step size $\alpha$ equals $\epsilon$, PGD reduces to FGSM, which can be seen as a special case of PGD. The iterative nature of PGD allows it to find more effective adversarial perturbations compared to FGSM, making it a stronger attack method used in adversarial training to enhance model robustness.

## A.3 UNIVERSAL ADVERSARIAL PERTURBATIONS (UAP)

Similar to Moosavi-Dezfooli et al. (2016), we aim to find perturbations that are state-agnostic, such that a single perturbation can be applied to all the images to cause failure of the agent. To this end, we collect few samples of state-action pairs by rolling out our policy and optimizing the perturbations as a parameter to minimize the loss similar to our PGD attacks.

---

**Algorithm 3** Computation of Universal Perturbations for Behavior Cloning

---

**Require:** Data points $\mathcal{D} = \{(s_i, a_i^{target})\}_{i=1}^N$, behavior cloning model $\pi_\theta$, desired $\ell_p$ norm of the perturbation $\xi$
**Ensure:** Universal perturbation vector $v$
1: Initialize $v \leftarrow 0$
2: **for** each datapoint $(s_i, a_i^{target}) \in \mathcal{D}$ **do**
    $a_i = \text{Algorithm}(s_i + v)$
    $v = v - \alpha \cdot \nabla_v Loss(a_i, a_i^{target})$
3: **end for**
4: **return** $v$

---

# B TASK DESCRIPTION

We evaluate the vulnerability of behavior cloning methods on several manipulation tasks of varying complexity. Each task is implemented in both simulation using MuJoCo and the robosuite framework, as well as on real Franka Emika Panda robots.

## B.1 LIFT

A foundational manipulation task where a robot arm must lift a small cube (4cm x 4cm x 4cm) from a table surface. The task tests basic pick-and-place capabilities and serves as an entry-level benchmark. Success is determined by elevating the cube above a threshold height. Initial cube poses are randomized with z-axis rotation within a small square region at the table center.

## B.2 CAN

A manipulation task requiring the robot to transfer a soda can from a large source bin into a smaller target bin. This task presents increased difficulty over Lift due to the more complex grasping requirements of the cylindrical can and the constrained placement target. The can's initial pose is randomized with z-axis rotation anywhere within the source bin.

## B.3 SQUARE

A high-precision manipulation task where the robot must pick up a square nut and insert it onto a vertical rod. This task significantly increases complexity by requiring precise alignment and complex insertion dynamics. The nut's initial pose is randomized with z-axis rotation within a square region on the table surface.

## B.4 PUSH-T

A contact-rich manipulation task adapted from (Florence et al., 2021) where the robot must guide a T-shaped block to a fixed target location using a circular end-effector. The task requires precise control of contact dynamics, as the robot must strategically apply point contacts to maneuver the block along the desired trajectory. Unlike pick-and-place tasks, success depends on understanding and exploiting the complex dynamics of planar pushing. We evaluate using RGB image observations augmented with end-effector proprioception. Initial positions of both the T-shaped block and the end-effector are randomized to ensure learned policies must generalize across different pushing strategies.

## B.5 TOOL HANG

It's the most difficult task in robomimic suite, as it requires a robotic arm to assemble the frame consisting of a base piece and hook piece by inserting the hook into the base, and hang a wrench on the hook. This task at multiple stages necessitates precise, and dexterous, rotation-heavy movements. Initial position of the insertion hook as well as that of ratcheting wrench and z-rotation are randomized in a small square at the beginning of the episode.

## C  BEHAVIOR CLONING POLICIES

### C.1  IMPLICIT BEHAVIOR CLONING

---

**Algorithm 4** Implicit BC Inference

---

**Require:** Trained energy model $E_\theta(s, a)$, observation $s$, number of samples $N_{samples}$, number of iterations $N_{iters}$, initial sampling std. dev. $\sigma_{init}$, decay rate $K$

1: Initialize $\{\tilde{a}^i\}_{i=1}^{N_{samples}} \sim \mathcal{U}(a_{min}, a_{max})$, $\sigma = \sigma_{init}$

2: **for** $iter = 1, 2, \ldots, N_{iters}$ **do**

3:  $\quad \{E_i\}_{i=1}^{N_{samples}} \leftarrow \{E_\theta(s, \tilde{a}^i)\}_{i=1}^{N_{samples}}$ $\hfill \triangleright$ Compute energies

4:  $\quad \{\tilde{p}_i\}_{i=1}^{N_{samples}} \leftarrow \left\{ \frac{e^{-E_i}}{\sum_{j=1}^{N_{samples}} e^{-E_j}} \right\}_{i=1}^{N_{samples}}$ $\hfill \triangleright$ Compute softmax probabilities

5:  $\quad$ **if** $iter < N_{iters}$ **then**

6:  $\quad\quad \{\tilde{a}^i\}_{i=1}^{N_{samples}} \leftarrow \text{Multinomial}(N_{samples}, \{\tilde{p}_i\}_{i=1}^{N_{samples}}, \{\tilde{a}^i\}_{i=1}^{N_{samples}})$ $\hfill \triangleright$ Resample with replacement

7:  $\quad\quad \{\tilde{a}^i\}_{i=1}^{N_{samples}} \leftarrow \{\tilde{a}^i + \mathcal{N}(0, \sigma)\}_{i=1}^{N_{samples}}$ $\hfill \triangleright$ Add noise

8:  $\quad\quad \{\tilde{a}^i\}_{i=1}^{N_{samples}} \leftarrow \text{clip}(\{\tilde{a}^i\}_{i=1}^{N_{samples}}, a_{min}, a_{max})$ $\hfill \triangleright$ Clip to bounds

9:  $\quad\quad \sigma \leftarrow K\sigma$ $\hfill \triangleright$ Shrink sampling scale

10:  $\quad$ **end if**

11: **end for**

12: $i = \arg\max_i(\{\tilde{p}_i\}_{i=1}^{N_{samples}})$

13: **return** $\tilde{a}^i$

---

### C.2  DIFFUSION POLICY

For Diffusion policy we use absolute positional actions as the original work shows that CNN-based diffusion policy performs poorly with robomimic's offical dataset, that uses veloocity control, as in the actions are represented as delta with respect to the current.

---

**Algorithm 5** Diffusion Policy Inference

---

**Require:** Observation horizon $T_0$, Action Horizon $T_a$, Prediction Horizon $T_p$, State sequence $\mathbf{S}_t = \{\mathbf{s}_{t-T_o+1}, \ldots, \mathbf{s}_t\}$, number of denoising iterations $K$

**Ensure:** Action sequence $\mathbf{A}_t = \{\mathbf{a}_t, \ldots, \mathbf{a}_{t+T_p-1}\}$

1: Initialize $\alpha, \gamma, \sigma$

2: Initialize $\mathbf{A}_t^{(K)} \sim \mathcal{N}(\mathbf{0}, \mathbf{I})$

3: **for** $k = K, K-1, \ldots, 1$ **do**

4:  $\quad \mathbf{A}_t^{(k-1)} = \alpha(\mathbf{A}_t^{(k)} - \gamma\epsilon_\theta(\mathbf{S}_t, \mathbf{A}_t^{(k)}, k)) + \sigma\mathcal{N}(\mathbf{0}, \mathbf{I})$

5: **end for**

6: **return** $\mathbf{S}_t$

---

## D  HYPERPARAMETERS

We adopt the following temporal horizons from Diffusion Policy:

- Action prediction horizon ($T_p$): 16 steps
- Action execution horizon ($T_a$): 8 steps
- Observation context window ($T_o$): 2 steps

For the adversarial attacks, we use the following settings:

1. Overall attack budget:

- $\varepsilon = 0.0625$ (16/256) $L_\infty$ norm (normalized to input range $[0, 1]$)
- Perturbations are clipped to $[0, 1]$ range

2. Framework-specific perturbation bounds:
   - For standard BC frameworks on Robomimic: $[0.15, 0.15, 0]$ in $(x, y, z)$ directions for relative end-effector positions
   - For Diffusion Policy on Robomimic: $[0.45, 0.45, 0]$ in $(x, y, z)$ directions for absolute end-effector positions. The larger perturbation magnitude accounts for the absolute position representation, compared to relative positions used in other frameworks
   - For all the frameworks on Push-T: $[100, 100]$ for the two action dimensions.

3. PGD attack parameters:
   - Number of iterations: 40
   - Per-iteration step size ($\varepsilon_{\text{iteration}}$): 0.005

For IBC inference, we use derivative-free optimization with $N_{\text{samples}} = 1024$.

## D.1 TARGET ACTION SELECTION

For targeted attacks across all algorithms, target actions are generated by perturbing the expected clean actions:

$$a_{target} = a_{clean} + \delta_{action} \tag{4}$$

where $a_{clean}$ is the action predicted by the unperturbed policy and $\delta_{action}$ is the desired action perturbation. For our experiments, we set $\delta_{action} = [0.15, 0.15]$ for perturbations in x and y directions for all frameworks except Diffusion Policy, where we use $\delta_{action} = [0.45, 0.45]$. These values were chosen to ensure the target actions remain within physically feasible bounds while being sufficiently different from the clean actions to potentially cause task failures. For PGD attacks, this target action computation is performed at each inference step using the current clean action prediction, while for UAP the target actions are computed once using the perturbed offline action trajectories.

## E RESULTS ON ADDITIONAL ENVIRONMENTS

### E.1 SQUARE ENVIRONMENT

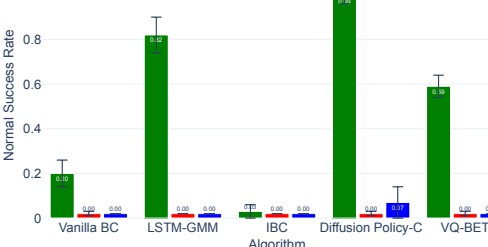
(a) PGD attacks for square.

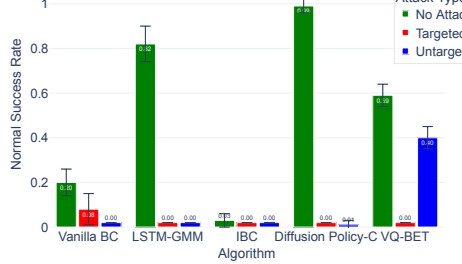
(b) Universal perturbation attacks for square.

Figure 5: Comparison of PGD and universal perturbation attacks for square task.

Table 5: Inter-Algorithm Transferability of Universal Untargeted Perturbations for Square

| Target Policy
Attacker | Square | | | | |
|---|---|---|---|---|---|
| | Vanilla BC | LSTM-GMM | IBC | DiffusionPolicy-C | VQ-BET |
| Random | 0.42 | 0.36 | 0.00 | 0.98 | 0.64 |
| Vanilla BC | 0.00 | 0.08 | 0.00 | 0.94 | 0.38 |
| LSTM-GMM | 0.18 | 0.00 | 0.00 | 0.96 | 0.62 |
| IBC | 0.42 | 0.1 | 0.00 | 0.98 | 0.62 |
| DiffusionPolicy-C | 0.00 | 0.00 | 0.00 | 0.00 | 0.32 |
| VQ-BET | 0.26 | 0.00 | 0.00 | 0.98 | 0.00 |

## E.2 CAN ENVIRONMENT

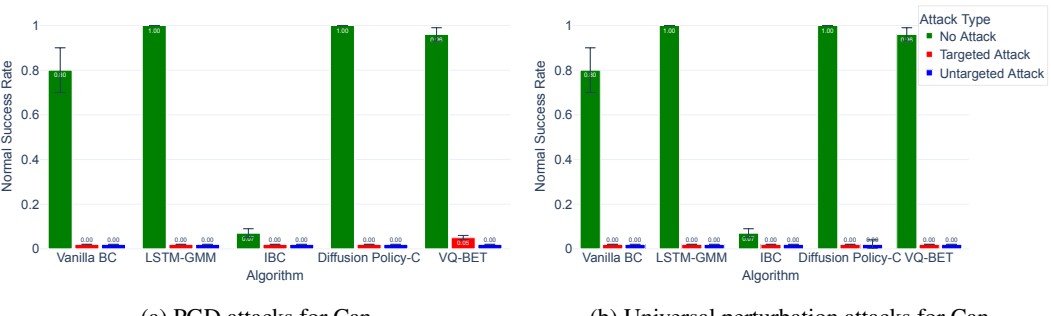

(a) PGD attacks for Can.   (b) Universal perturbation attacks for Can.

Figure 6: Comparison of PGD and universal perturbation attacks for Can task.

Table 6: Inter-Algorithm Transferability of Universal Untargeted Perturbations for Can

| Target Policy
Attacker | Can | | | | |
|---|---|---|---|---|---|
| | Vanilla BC | LSTM-GMM | IBC | DiffusionPolicy-C | VQ-BET |
| Random | 0.62 | 0.94 | 0.00 | 1.00 | 0.96 |
| Vanilla BC | 0.00 | 0.66 | 0.00 | 0.42 | 0.88 |
| LSTM-GMM | 0.18 | 0.00 | 0.00 | 0.72 | 0.68 |
| IBC | 0.72 | 0.98 | 0.00 | 1.00 | 0.92 |
| DiffusionPolicy-C | 0.02 | 0.24 | 0.00 | 0.00 | 0.70 |
| VQ-BET | 0.34 | 0.64 | 0.00 | 0.42 | 0.04 |

Table 7: Inter-Architecture Transferability. Transferability of attacks trained with resnet-18 to resnet-50 as backbone for Can.

| Algorithm | NA Resnet-18 | NA Resnet-50 | Resnet-18 | Resnet-50 |
|---|---|---|---|---|
| Vanilla BC | 0.75 | 0.70 | 0.00 | 0.34 |
| LSTM-GMM | 1.00 | 0.25 | 0.00 | 0.00 |
| IBC | 0.09 | 0.00 | 0.00 | 0.00 |
| DiffusionPolicy-C | 1.00 | 0.875 | 0.00 | 0.30 |
| VQ-BET | 1.00 | 0.70 | 0.04 | 0.70 |

## F RANDOMIZED SMOOTHING

Randomized smoothing is a technique used to enhance the robustness of deep neural networks against adversarial perturbations. The core idea is to smooth the model's predictions by averag-

ing over multiple randomly perturbed versions of the input. For a given input state $s$, the smoothed policy $\tilde{\pi}(s)$ is defined as:

$$\tilde{\pi}(s) = \mathbb{E}_\varepsilon[\pi(s + \varepsilon)], \quad \text{where} \quad \varepsilon \sim \mathcal{N}(0, \sigma^2 I) \tag{5}$$

During inference, we approximate this expectation by averaging predictions over $N$ randomly sampled perturbations:

$$\tilde{\pi}(s) \approx \frac{1}{N} \sum_{i=1}^{N} \pi(s + \varepsilon_i), \quad \text{where} \quad \varepsilon_i \sim \mathcal{N}(0, \sigma^2 I) \tag{6}$$

### F.1 IMPLEMENTATION DETAILS

For our experiments, we used:

- Number of random samples ($N$): 100
- Noise standard deviation ($\sigma$):
    - Lift task: $\sigma = 0.1$
    - Push-T task: $\sigma = 0.05$

The $\sigma$ values were carefully chosen through validation to maintain performance on clean (non-attacked) inputs while providing meaningful defense against adversarial perturbations.

### F.2 RESULTS AND ANALYSIS

As shown in Tables 8 and 9, randomized smoothing demonstrates varying degrees of effectiveness across different algorithms and tasks:

**Lift Task Results:**

- Diffusion Policy shows the most impressive improvement, with task success rate improving significantly (from 25% to 98% failure under PGD attacks)
- VQ-BET and Vanilla BC show moderate improvements (8% to 66% and 48% to 52% respectively)
- IBC demonstrates a notable improvement from 21% to 50% task success rate
- LSTM-GMM shows limited benefit from smoothing

**Push-T Task Results:**

- The benefits of randomized smoothing are less pronounced in Push-T task
- IBC shows the most significant improvement (from 38% to 50% task success rate)
- Other algorithms show minimal improvements, this could partly be because of multi-modal nature of the data distribution in the PushT environment (Lee et al., 2024), where averaging individual predictions might lead to the mean between them.

The difference in effectiveness between tasks suggests that randomized smoothing's utility may be task-dependent, with simpler manipulation tasks with no multi-modality benefiting more from this defense strategy than tasks that are inherently multi-modal.

Table 8: Comparison of the randomized smoothing on the algorithmic performance for the lift task.

| Algorithm | NA | NA Randomized Smoothing | PGD Attack | Randomized Smoothing with PGD |
|---|---|---|---|---|
| Vanilla BC | 1.00 | 1.00 | 0.48 | 0.52 |
| LSTM-GMM | 1.00 | 0.93 | 0.00 | 0.00 |
| IBC | 0.95 | 0.80 | 0.21 | 0.50 |
| DiffusionPolicy-C | 1.00 | 1.00 | 0.25 | 0.98 |
| VQ-BET | 1.00 | 1.00 | 0.08 | 0.66 |

Table 9: Comparison of the randomized smoothing on the algorithmic performance for the Pusht task.

| Algorithm | NA | NA Randomized Smoothing | PGD Attack | Randomized Smoothing with PGD |
|---|---|---|---|---|
| Vanilla BC | 0.74 | 0.74 | 0.08 | 0.08 |
| LSTM-GMM | 0.66 | 0.54 | 0.00 | 0.00 |
| IBC | 0.68 | 0.67 | 0.38 | 0.50 |
| DiffusionPolicy-C | 0.88 | 0.84 | 0.23 | 0.24 |
| VQ-BET | 0.72 | 0.71 | 0.10 | 0.10 |

## G  INTER-TASK TRANSERABILITY

We investigate the transferability of untargeted Adversarial Perturbation attacks developed in one environment to unseen new environments. We use the attacks developed for the Lift task across all algorithms and measure their ability to impact performance of the respective algorithms in both the Can and Square tasks. For every task, we report the percentage decrease in the robot task completion rate compared to the non-attacked version.

Our results in Table 10 show that the attacks developed in Lift can transfer to both the other environments, often decreasing the performance of the attacked policy. However in rare case of BC for Square, we see an unexpeccted increase in performance when attacked using the attack developed for Lift envionrment but this could be due to random initialization of environments and the time constraint for testing only 3 seeds for each algorithm. It could also be due to the fact that adding a small amount of action noise to policies can sometimes increase performance by helping the policy get unstuck.

Table 10: Multi-Task Transferability of the Universal Pertubations

| Algorithm \ Task | LIFT | CAN | LIFT-TO-CAN | SQUARE | LIFT-TO-SQUARE |
|---|---|---|---|---|---|
| Vanilla BC | 100% | 100% | 50% | 100% | -40% |
| LSTM-GMM | 100% | 100% | 40% | 100% | 38.9% |
| IBC[*] | 7.89% | 100% | 100% | 100% | 100% |
| DiffusionPolicy-C | 100% | 100% | 45% | 100% | 6.6% |
| VQ-BET | 100% | 100% | 0% | 100% | 11.4% |

[*]IBC has very low (almost zero) performance on the Can and Square task, so the above metric may not capture the full picture for (only) IBC.

## H  ILLUSTRATIONS

In this section, we show examples of the adversarial perturbations. Figure 7 shows an example of *untargeted attacks* on the visual input for the Lift task. Figure 8 shows an example of *targeted attacks* on the visual input for the Lift task. Figure 9 shows an example of *untargeted attacks* on the visual input for the Push-T task. Figure 10 shows an example of *targeted attacks* on the visual input for the Push-T task. We note that these perturbations are minor and in some cases almost imperceptible.

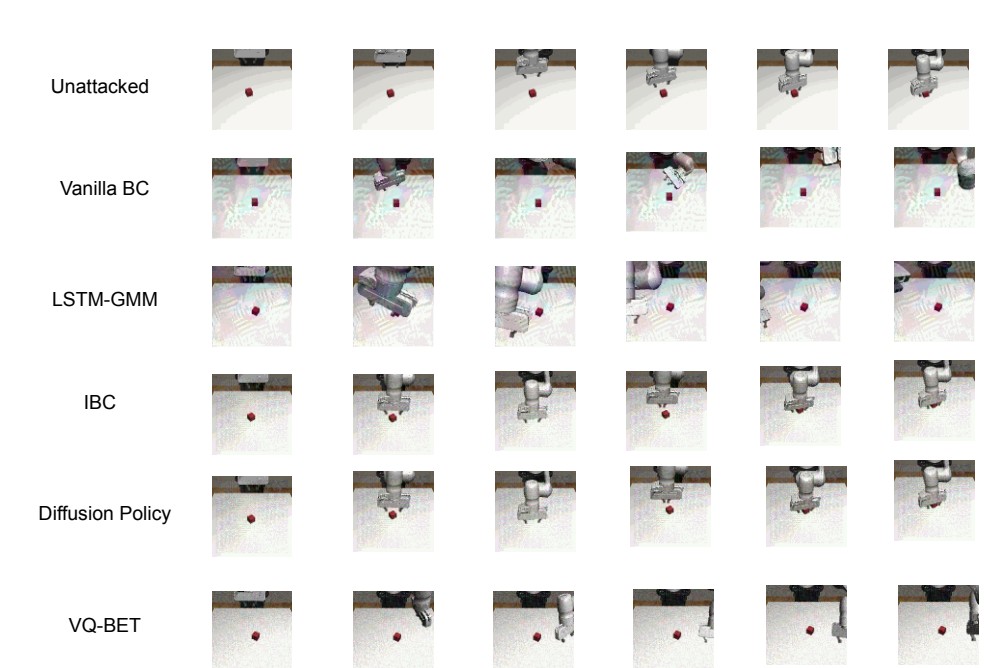

Figure 7: Untargeted Attacks on Lift task.

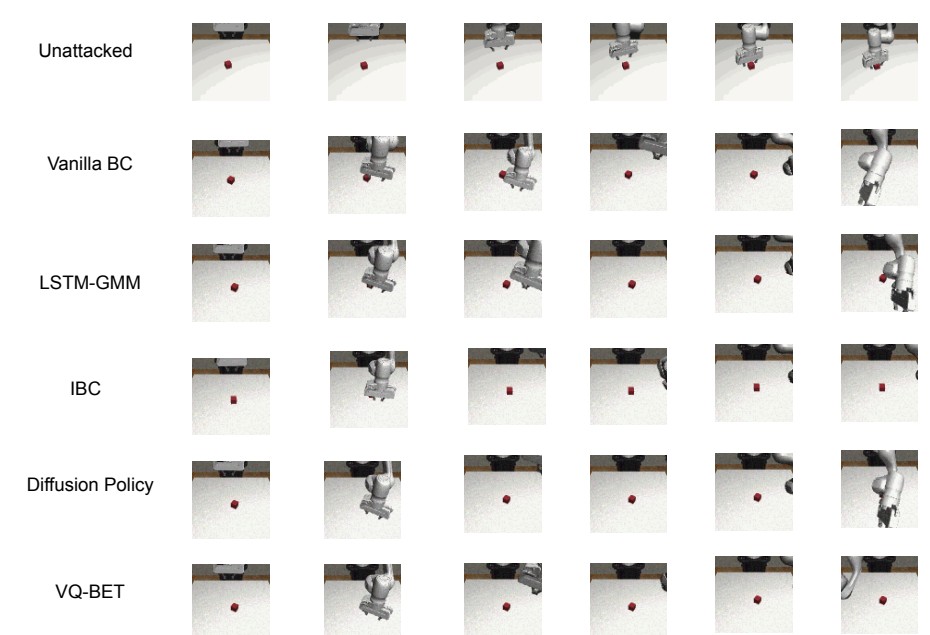

Figure 8: Targeted Attacks on Lift task, where the target direction is towards top-left corner of the object.

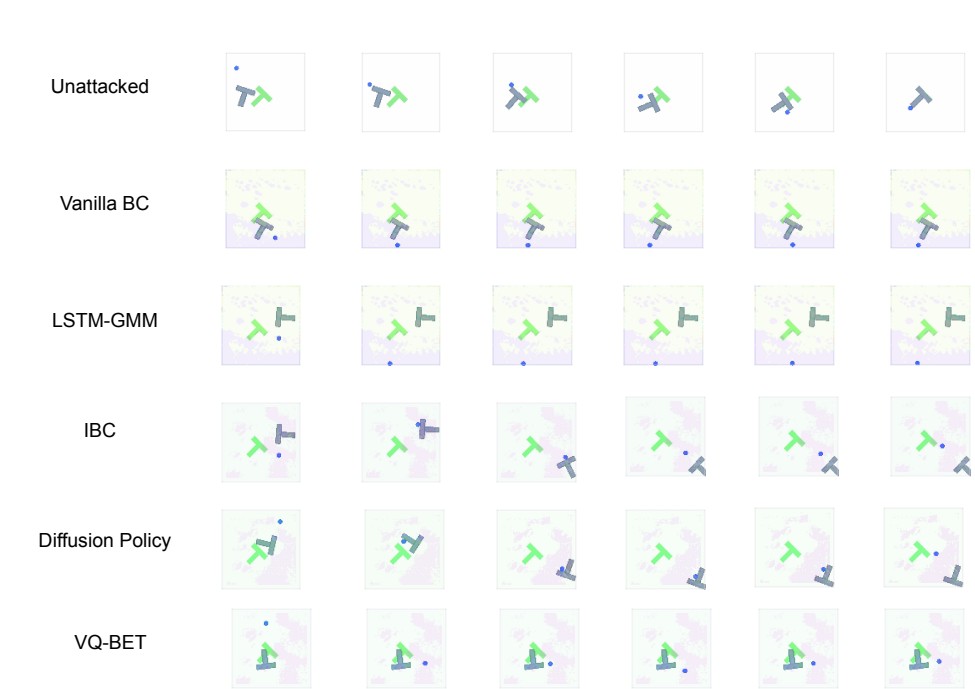

Figure 9: Untargeted Attacks on PushT task.

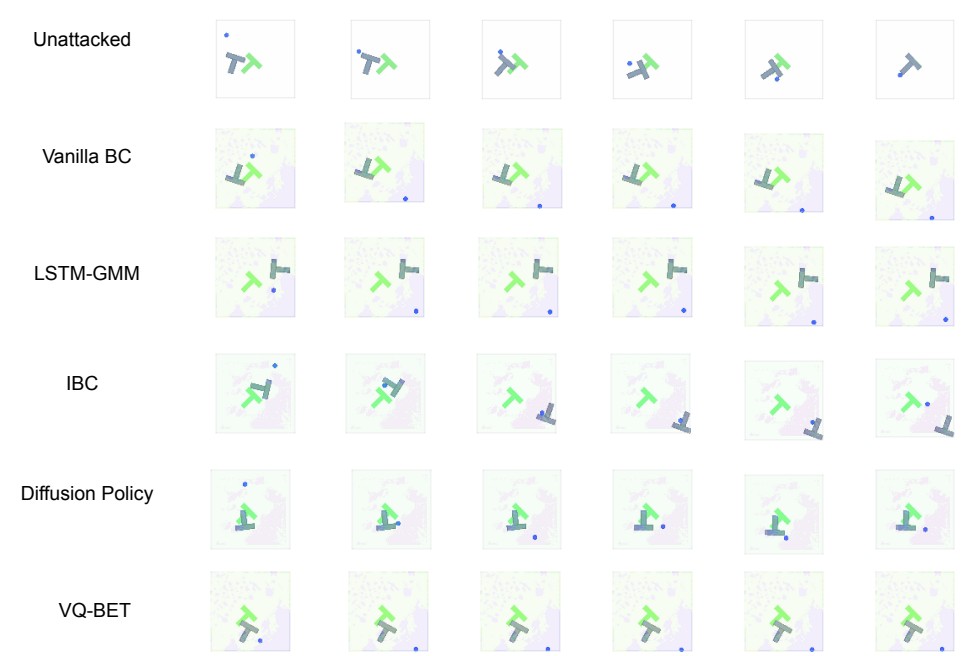

Figure 10: Targeted Attacks on PushT task, where the target is bottom right corner of the environment.

## I   TOOL HANG

We investigate the vulnerability of modern behavior cloning algorithms on Tool Hang, specifically we look at the targeted universal perturbation attack for Diffusion Policy, LSTM-GMM, IBC and Vanilla BC. As before, we use pre-trained checkpoints for Diffussion Policy , LSTM-GMM and IBC. We train our own policies for Vanilla-BC. Training VQ-BET on this task is extremely slow and due to time constraints during the rebuttal phase we couldn't finish training VQBET policies all 3 seeds but we promise to have these and their attacked versions by camera-ready deadline. We report mean and standard deviation of success rate across 3 different seeds where we evaluate each seed policy for 50 randomly initialized environments. Our results in Fig 11 show a similar trend as in other tasks and environments, of decrease in performance of the attacked policy for all algorithms except BC and IBC which fail to even learn a good behavior cloned policy owing to difficulty of the task. As observed before, Diffusion Policy seems to be more robust than LSTM-GMM to the universal pertubatioon attack.

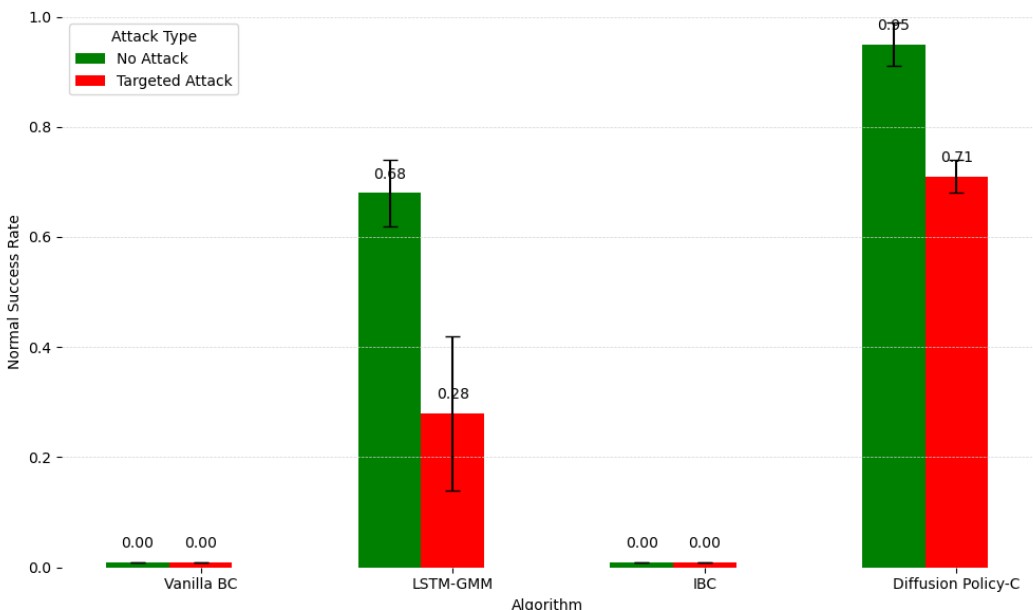

Figure 11: UAP for Tool Hang task. The y axis denotes the normal performance of the evaluated policies, which is the lower the better for attacks.0.65

## J   SENSITIVITY TO EPSILON VALUES

Our analysis reveals surprising vulnerabilities in behavior cloning algorithms even with minimal perturbations (for Universal Untargeted Attacks). As shown in Figure 12, while decreasing epsilon values generally reduces attack efficacy, algorithms like VQ-BET, LSTM-GMM, and Diffusion Policy still exhibit substantial performance degradation even at very small epsilon values ($\epsilon$ of 4/256). This heightened sensitivity to small perturbations highlights a concerning vulnerability in current behavior cloning approaches, suggesting that even well-constrained adversarial attacks can significantly compromise policy performance.

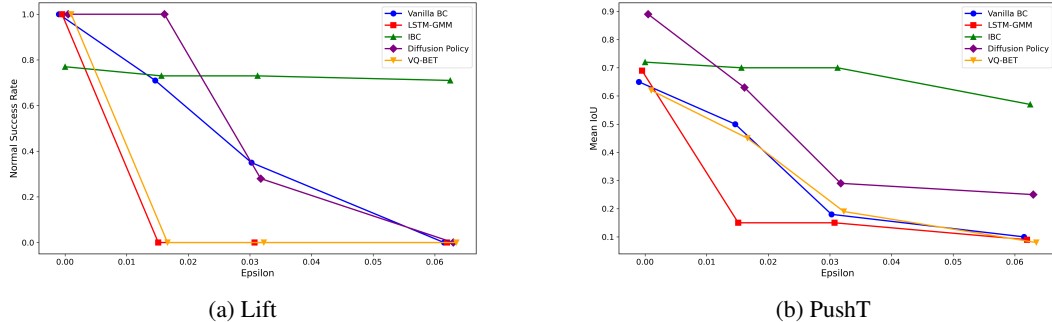

(a) Lift             (b) PushT

Figure 12: Performance of the algorithms to smaller epsilon values highlight the vulnerability and lack of robustness of the Behavior Cloning Algorithms.

