# OpenReview forum: "How vulnerable is my learned policy? Adversarial attacks on modern behavioral cloning policies"
_ICLR.cc/2025/Conference — Submitted to ICLR 2025_

### Official Review · Reviewer_9PBk · 2024-10-31

**Soundness:** 2
**Presentation:** 3
**Contribution:** 2
**Rating:** 5
**Confidence:** 4

**Summary:**

This paper presents a thorough study about the adversarial vulnerability of several representative imitation learning methods (e.g., Vanilla BC, LSTM-GMM, IBC, Diffusion Policy and VQ-BeT) under two white-box attacks (PGD and UAP). The evaluation also covers the transferbility of perturbations across different policies and architectures.

**Strengths:**

1. The transferbility of perturbation between policies and architectures is also investigated.
2. The paper is well organized and easy to follow.

**Weaknesses:**

1. I concur with the authors that naive success rates may not accurately reflect the impact of adversarial attacks on robotic policies. It is essential to develop additional metrics, particularly those addressing safety concerns associated with real-robot deployments.
2. The current version seems that we have to develop tailored attacks given specific task, with privilege knowledge about the model parameters (PGD) and training samples (UAP). Thus the practical implications are limited for real-world deployments.

**[Minor:]**
1. Subscripts are not used correctly in line 752-753

**Questions:**

1. It is noted that the perturbations are applied uniformly throughout the trajectory during inference concerning UAP attack. Then what about PGD? Are perturbations computed for every single step during inference with updated observations?
2. How to determine the 'target action' when performing targetted attack? Does it also need to be updated during the inference process?
3. What is the precise number of iterations or steps utilized in the PGD attack? It is my belief that an extensive optimization process may diminish the practicality of the attack, as it could become less 'stealthy' for robots engaged in real-time operations.
4. As indicated in Tables 5 and 6, the IBC policy appears to be more vulnerable in the <Square> and <Can> tasks (than <Lift> and <Push-T>), while the LSTM-GMM demonstrates exceptional robustness in the <Can> task. Do these results imply that it is challenging to draw definitive conclusions regarding the relative robustness of the different policies (models), given that their performance varies across tasks?  Could the authors discuss potential factors that might contribute to the varying performance?
5. Have the authors studied about multi-task learning setting? Do you expect perturbations could be transferred across tasks?

---

> ### Author Response · Authors · 2024-11-24
> **Response to reviewer 9PBk [Part 1]**
>
> Thank you very much for your helpful comments. We are happy that the reviewer found our paper to be well organized and easy to follow, as well acknowledge our transferability studies.
>
>
> > I concur with the authors that naive success rates may not accurately reflect the impact of adversarial attacks on robotic policies. It is essential to develop additional metrics, particularly those addressing safety concerns associated with real-robot deployments.
>
> We used robot task success since it is the primary metric used in prior behavior cloning policy evaluation. While success is one very important metric, we agree that there are other metrics that could be considered in the future such as collisions or other constraint violations, human safety and comfort in human-robot collaborative tasks, etc. We agree that this is an exciting area of future work, but note that the best safety metrics are not obvious and are probably very task dependent, thus we focus on the standard task success rate in this paper.
>
> > The paper develops tailored attacks given a specific task, with privileged knowledge about the model parameters (PGD) and training samples (UAP). Thus the practical implications are limited for real-world deployments.
>
> Given the increasing trend toward open-sourced ML models, we believe the types of attacks we study are a very practical concern that has not received attention in prior work.  Many practitioners and developers still fully understand or appreciate the vulnerability of the algorithms they design and deploy. We are the first to showcase this vulnerability across modern behavior cloned methods. We also note that our paper investigates the transferability of different attacks across algorithms, vision backbones, and tasks. These transfer attacks remove the need to know the specific model parameters, training samples, or task. We also note that early work on adversarial attacks in computer vision focused primarily on white-box attacks which led to a large amount of interest and different lines of research. In a similar vein, we believe our paper will lead to an increased interest in the vulnerability of behavior cloned policies and provide the first step along an exciting new research area.
>
> > Subscripts are not used correctly in line 752-753
>
> Good catch. These have been fixed.
>
> > It is noted that the perturbations are applied uniformly throughout the trajectory during inference concerning UAP attack. Then what about PGD? Are perturbations computed for every single step during inference with updated observations?
>
> Yes. Universal Perturbations refers to an offline attack where the perturbation is calculated for the task on the dataset and applied at the time of inference without changing it based on observations. However, PGD is an online attack, where the attacks adapts the perturbation for every step based on the observation. We have clarified this in Section 3.1 Threat Model.
>
> > How to determine the 'target action' when performing a targetted attack? Does it also need to be updated during the inference process?
>
> The target action for targeted attacks is computed by adding a desired perturbation to the expected (clean) action for all algorithms. Specifically, if a_clean is the action predicted by the policy without any adversarial perturbation, the target action is calculated as:
> a_target = a_clean + δ_action
> where δ_action represents the desired perturbation to the action. This approach ensures the target actions remain within a feasible range while still causing potential task failures. We have clarified this in Appendix D.1.
>
> > What is the precise number of iterations or steps utilized in the PGD attack? It is my belief that an extensive optimization process may diminish the practicality of the attack, as it could become less 'stealthy' for robots engaged in real-time operations.
>
> We have used 40 steps for all the PGD attacks. We have added this in the Hyperparameters section of the paper in Appendix D.

---

> > ### Author Response · Authors · 2024-11-24
> > **Response to reviewer 9PBk [Part 2]**
> >
> > > As indicated in Tables 5 and 6, the IBC policy appears to be more vulnerable in the <Square> and <Can> tasks (than <Lift> and <Push-T>), while the LSTM-GMM demonstrates exceptional robustness in the <Can> task. Do these results imply that it is challenging to draw definitive conclusions regarding the relative robustness of the different policies (models), given that their performance varies across tasks? Could the authors discuss potential factors that might contribute to the varying performance?
> >
> > Great question. We would like to emphasize that Tables 5 and 6 show the inter-algorithm transferability of universal adversarial perturbations rather than direct robustness measurements. We would like to clarify several important points:
> >
> > **Baseline Performance**: As shown in Figures 5 and 6, IBC's base performance (before any attacks) is notably low for both Square (0.03) and Can (0.07) tasks. This means we cannot meaningfully compare its robustness to attacks when the policy isn't successfully learning these tasks in the first place.
> >
> > **Random Noise Sensitivity**: Looking at Tables 5 and 6, random noise perturbations are sufficient to reduce IBC's success rate to 0 for both tasks. This suggests fundamental issues with the base policy rather than specific vulnerabilities to adversarial attacks.
> >
> > **LSTM-GMM's Apparent Robustness**: While LSTM-GMM does show resistance to transferred perturbations in the Can task, this likely indicates that it learns significantly different feature representations compared to other algorithms rather than inherent robustness. This aligns with the paper's broader finding that "our results provide evidence that the different algorithms and the same algorithm trained with a different architecture are learning some similar features that are not completely orthogonal but also not completely similar" (Section 5).
> >
> > > Have the authors studied about multi-task learning setting? Do you expect perturbations could be transferred across tasks?
> >
> > Thank you for this great suggestion. We have added new experiments to our paper in Appendix G. We investigate the transferability of attacks developed in Lift task across all algorithms and measure their ability to impact performance of the respective algorithms in both Can and Square task . For every task, we report the percentage decrease in the task-success rate performance compared to the non-attacked version. We find that attacks in Lift often degrade performance of a policy when deployed in other environments. We think this is a very exciting result that shows that attacks can generalize beyond the task they were optimized for. We hope this will lead to more investigation in future work and agree that it makes our paper stronger and more interesting.

---

> ### Comment · Area_Chair_TZUD · 2024-11-27
> **Rebuttal Response**
>
> Dear Reviewer,
> Do you mind letting the authors know if their rebuttal has addressed your concerns and questions? Thanks!
> -AC

---

> ### Comment · Reviewer_9PBk · 2024-11-27
> **Thanks for the efforts made on the rebuttal**
>
> The authors' response addressed most of my concerns. But I think this paper in its current form still poses several limitations:
>
> 1. **From the perspective of how to build adversarial attacks against imitation learning policies:** The white-box assumptions of UAP (Universal Adversarial Perturbation) and PGD (Projected Gradient Descent) significantly limit the feasibility of attacks in real-world systems. Especially for PGD, which, as mentioned by the authors, is implemented with a perturbation budget of about $\epsilon=16/255$ (presumably under the $L_{\infty}$ norm, this is also not clearly noted in the revised paper), and optimization steps of 40. PGD necessitates online updates for each inference step. Given that a policy would require an additional 40 rounds of forward-backward passes before executing an action, this further reduces its potential threat to real-world applications.
>
> 2. **From the perspective of a comprehensive study of the vulnerability of current IL policies:** It's rather hard to draw apprent conclusions from the evaluations conducted. However, I believe vulnerability of policies would be an interesting direction to explore. A more in-depth analysis is necessary to enhance the potential impact of the paper.
>
> I appreciate the overall improvements and the effort made during the rebuttal process. While I would like to increase my score, I still lean toward recommending rejection.

---

> > ### Author Response · Authors · 2024-11-27
> > **Response to Reviewer 9P8k**
> >
> > Thank you very much for your response. In order to further clarify your concerns:
> >
> > 1. We have included the hyperparameter ablation for the $\epsilon = 0, 4/256, 8/256, 16/256$ for the PushT and Lift environments in the Appendix J. While we agree that PGD attack might not be the most feasible attack in real-world systems, but  since there were no previous studies in this direction we felt the necessity to include to have an upper-bound of what is achievable in case of white-box access and per-step attack. However, Universal Attacks are much more feasible and present a real threat, in addition to the real threat we demonstrate of transferability of the attack across algorithms and tasks, and implies that a potential adversary does not need white-box access to a particular algorithm. Thus, our paper demonstrates both worst-case vulnerability and more pragmatic vulnerability. We believe both are extremely important and that our paper is a valuable addition to the literature and will stimulate a lot of interest and future
> > work.
> >
> > 2. We would like to emphasize that our paper does provide the first comprehensive study of the vulnerability of modern IL policies. We agree that this is an interesting direction and we argue that most members of the ICLR community and ML community at large will also find our results interesting.
> >
> > - First, our paper reveals that modern behavior cloning algorithms, particularly VQ-BET, are surprisingly vulnerable to small adversarial perturbations, with even epsilon values as small as 4/256 causing significant performance degradation (Line 412-413, Fig 12 Appendix J).
> >
> > - Second, we discover that implicit policies (IBC and Diffusion Policy) show greater robustness compared to explicit policies, likely due to their stochastic action selection process.
> >
> > - Third, we demonstrate interesting transferability patterns - attacks can transfer between algorithms and even across different tasks, though effectiveness varies with task complexity. A particularly intriguing finding is that Diffusion Policy's robustness increases with larger action horizons, suggesting a trade-off between prediction length and vulnerability.
> >
> > - Finally, as suggested by the Reviewer DYsQ, we uncover that traditional computer vision defense methods like randomized smoothing (Line 536, Appendix F) are less effective for multi-modal action distributions, highlighting unique challenges in defending robotic policies against adversarial attacks.
> >
> >  We are a bit confused by this comment due to the fact that we strongly believe we already provide the in-depth analysis requested. Is there any particular experiment you believe is missing that would strengthen our paper?

---

> > ### Author Response · Authors · 2024-12-01
> > **Reply to Reviewer 9P8k**
> >
> > Dear Reviewer, while PGD has computational overhead, our universal perturbation and transferability results demonstrate practical vulnerabilities without requiring online computation or white-box access. Combined with our thorough analysis showing vulnerability even with small ε values (4/256) and novel insights about implicit vs explicit policies, would you consider raising your score?

---

### Official Review · Reviewer_DYsQ · 2024-11-02

**Soundness:** 2
**Presentation:** 3
**Contribution:** 2
**Rating:** 5
**Confidence:** 4

**Summary:**

The paper studies the vulnerability of commonly used behavioral cloning algorithms. It considers well-known PGD and UVA attacks developed for supervised learning and directly applies them to compromise Behavior Cloning (BC), LSTM-GMM, and VQ-Behavior Transformer (VQ-BET) algorithms. It also shows that the PGD attack can be adapted to compromise Implicit Behavior Cloning (IBC) and Diffusion Policy (DP).

**Strengths:**

The paper highlights that commonly used behavioral cloning algorithms are vulnerable to PGD attacks.

**Weaknesses:**

The threat model is undefined. Are we considering data poisoning attacks? What can the attacker modify, states, actions, or both? What is the attacker's goal? What are the constraints to the attacker? None of them are defined in the paper. From 3.2.1, it seems that the paper considers both targeted and untargeted attacks. But the accurate definitions are missing, and it is unclear what tilde{p}_theta is.

While undefined in the paper, a reasonable attack objective is to induce the agent to learn a bad policy. In this case, the attacker should modify multiple data points collectively. However, the simple attacks considered in the paper, such as PGD and UAP, were developed for the one-shot supervised learning setting and are ineffective for the sequential setting. The new attacks for IBC and Diffusion Policy are straightforward adaptions of PGD and also myopic.

The reason why the simple myopic attacks can still work, as shown in the paper, is because the paper completely ignores defenses. However, there are well-known defenses for supervised learning, such as adversarial training and randomized smoothing, that can be easily adapted to the sequential setting when attacks are myopic, as considered in the paper. Simply showing that unprotected behavioral cloning algorithms are vulnerable to PGD attacks is not very interesting.

**Questions:**

Please see the discussion on weaknesses above.

---

> ### Author Response · Authors · 2024-11-24
> **Response to reviewer DYsQ**
>
> > The threat model is undefined. Are we considering data poisoning attacks? What can the attacker modify, states, actions, or both? What is the attacker's goal? What are the constraints to the attacker? None of them are defined in the paper. .
>
> Thank you for pointing this out. We would like to clarify that we are not considering data poisoning attacks. We focus on white-box, post-deployment attacks. We have added text to clarify this in our introduction (paragraph 5) and Section 3.1 to clarify this and describe our threat model. Unlike training time attacks that corrupt the learning process, we focus on adversarial attacks on open-sourced, pretrained models. Given the increasing trend toward open-sourced ML models, we believe the types of attacks we study are a very practical concern that has not received attention in prior work.
>
> > From 3.2.1, it seems that the paper considers both targeted and untargeted attacks. But the accurate definitions are missing, and it is unclear what tilde{p}_theta is.
>
> Thank you very much! We have clarified this.
>
> > While undefined in the paper, a reasonable attack objective is to induce the agent to learn a bad policy. In this case, the attacker should modify multiple data points collectively. However, the simple attacks considered in the paper, such as PGD and UAP, were developed for the one-shot supervised learning setting and are ineffective for the sequential setting. The new attacks for IBC and Diffusion Policy are straightforward adaptions of PGD and also myopic.
>
> While we agree that attacks that result in learning bad policy are important to study, we would like to emphasize that these attacks are out of scope of our paper. As mentioned above, we focus on post-training attacks on policies already trained to be good. Thus, we are not trying to get the agent to learn bad behavior. Furthermore, our results show that our PGD and UAP attacks are very successful (result in large drops in robot task performance)  in the sequential settings we consider.
>
>
> > The reason why the simple myopic attacks can still work, as shown in the paper, is because the paper completely ignores defenses. However, there are well-known defenses for supervised learning, such as adversarial training and randomized smoothing, that can be easily adapted to the sequential setting when attacks are myopic, as considered in the paper.
>
> Thank you for this suggestion. As recommended, we ran additional trials using randomized smoothing in Appendix F. We have observed that while randomized smoothing can definitely help achieve some amount of adversarial robustness, it does come at the cost of increased response time for dynamic problems such as robotics. We also want to highlight that robotics is unique in the situation that the action distribution can have multiple modes, thus averaging the actions might lead to the mean of the modes which might be undesirable thus raising another security concern for robotics. While Adversarial Training is interesting to study in this context, we believe that currently it is out of scope of our paper and leave it as future work.
>
> > Simply showing that unprotected behavioral cloning algorithms are vulnerable to PGD attacks is not very interesting.
>
> Given the increasing trend toward open-sourced ML models, we believe the types of attacks we study are a very practical concern that has not received attention in prior work.  Many ML practitioners and developers still fully understand or appreciate the vulnerability of the algorithms they design and deploy. We are the first to showcase this vulnerability across modern BC methods. We note that image-conditioned policies are a very different class of algorithms than standard end-to-end classification networks, especially the implicit models we study such as IBC and DP. Our work also shows their vulnerability and we think it will lead to a larger interest in making these algorithms more robust.

---

> ### Comment · Reviewer_DYsQ · 2024-11-25
>
> I would like to thank the authors for the clarification, which has addressed some of my concerns. However, I am still not convinced that it is sufficient to consider myopic attacks such as PGD and show it is effective when there is no defense or weak defense. Given the large body of recent work on post-training state perturbation attacks and both pre-training and post-training defenses for general RL, where the problem studied by the paper is a special case, the contribution of the paper is rather limited.
>
> [1] Huan Zhang et al., Robust Deep Reinforcement Learning against Adversarial Perturbations on State Observations, NeurIPS 2020.
>
> [2] Yanchao Sun, et al., Who is the strongest enemy? Towards optimal and efficient evasion attacks in deep RL, ICLR 2022.
>
> [3] Yongyuan Liang et al., Efficient adversarial training without attacking: Worst-case-aware robust reinforcement learning, NeurIPS 2022.
>
> [4] Zhihe YANG and Yunjian Xu. DMBP: Diffusion model-based predictor for robust offline reinforcement learning against state observation perturbations, ICLR 2024.

---

> > ### Author Response · Authors · 2024-11-28
> > **Response to reviewer DYsQ**
> >
> > Thank you for your response. We would argue that vulnerability to even myopic attacks is a big concern and our results will be of great interest to both the ML and robotics communities as the policies we study are some of the most performant and deployable policies for complex robotic tasks, yet no one has studied their vulnerability before.
> > We would also like to note that the policies we consider are not just a special case of prior work. Implicit policies like implicit behavior cloning and diffusion policy are very different from policies obtained from RL. We also note that it is behavior cloned policies that are currently seeing the most excitement and deployment, not RL policies. Thus, we again emphasize the importance of studying and showcasing the vulnerability of these policies.
> >
> > We agree that defenses are important and have added randomized smoothing defenses to our paper. We do not agree with the statement that this is a weak defense. When learning from demonstrations, data is often very scarce and limits the effectiveness of adversarial training (Schmidt et al. "Adversarially robust generalization requires more data." NeurIPS 2018). Also in RL adversarial training doesn’t always help and can hurt performance (reference [1] suggested by the reviewer). Because we focus our paper on pretrained behavior cloned policies, we believe  randomized smoothing makes the most sense as a strong defense. We appreciate the reviewers suggestion to add results on randomized smoothing and agree that they strengthen our paper. Interestingly, our result show this defense has limited protection, providing a further reason our results will be of interest to the community and why we think our paper provides a strong contribution.
> >
> > Thank you for your references. Below we discuss each paper:
> >
> > > 1] Huan Zhang et al., Robust Deep Reinforcement Learning against Adversarial Perturbations on State Observations, NeurIPS 2020.
> >
> > This paper shows that adversarial training hurts the performance of deep RL and proposes a policy regularization technique during training for DRL that improves robustness. Extending this idea to enable robust regularization for the behavioral cloning setting we consider is an interesting area of future work, but does not appear trivial.
> > > [2] Yanchao Sun, et al., Who is the strongest enemy? Towards optimal and efficient evasion attacks in deep RL, ICLR 2022.
> >
> > This paper proposes a new adversarial attack for RL algorithms that seeks to find the optimal attack. Notably this paper only considers attacks, as our work does, yet was published at ICLR and has been well cited. Optimal attacks on imitation policies is an interesting direction of future work, but may not always be feasible or computationally tractable.
> > > [3] Yongyuan Liang et al., Efficient adversarial training without attacking: Worst-case-aware robust reinforcement learning, NeurIPS 2022.
> >
> > This is another interesting work based on worst-attack bellman updates that depend on Q-values and an attack critic which we do not have in the imitation learning setting since we assume no access to a reward function. Furthermore, this paper considers a very RL-centric adversarial training approach that is used during RL. Thus, this defense is not applicable to our setting where Q-values and rewards are unknown and we assume the imitation policy is already trained. We also note that this paper does not consider defenses for MDPs with both continuous action and visual observations like we do.
> > > [4] Zhihe YANG and Yunjian Xu. DMBP: Diffusion model-based predictor for robust offline reinforcement learning against state observation perturbations, ICLR 2024.
> >
> > This paper proposes an interesting idea of using a denoising diffusion process to make offline RL more robust to state perturbations. While this is an interesting approach, it appears very computationally intensive as the diffusion must be trained and then applied to every state. We note that this paper does not consider any MDPs with visual, pixel-based states so it is unclear how well the proposed approach scales. We also note that this paper only shows robustness to randomized-sampling noise attacks and it is unclear how robust it would be against the types of gradient-based adversarial attacks we consider.
> >
> > We agree that these are all relevant references and will add them to our related work. However, in the context of these references, we believe our paper still provides a strong contribution in terms of studying vulnerability and also showing that randomized smoothing is not sufficient to protect against adversarial attacks on imitation policies. While some of the above papers discuss interesting defenses, it is unclear how to extend them to our setting and provide very interesting avenues for future work.

---

> > ### Author Response · Authors · 2024-12-01
> > **Reply to reviewer DYsQ**
> >
> > Dear Reviewer, our paper provides several strong contributions beyond vulnerability to myopic attacks - including the first comprehensive analysis of modern BC methods (especially implicit policies), novel robustness patterns, important transferability results, and limitations of randomized smoothing for multi-modal distributions. Given these contributions and that BC policies (not RL) are seeing the most real-world deployment, would you consider raising your score?

---

> > > ### Comment · Reviewer_DYsQ · 2024-12-02
> > >
> > > I'd like to thank the authors for further clarifications and have raised my score.
> > >
> > > However, my main concern is still that neither the proposed attack methods nor the newly added defense provide fundamentally new ideas or insights to the adversarial machine learning community. The attacks were directly adapted from well-known myopic attacks such as PGD, which were known to be insufficient to compromise robust RL methods with strong defenses applied, as shown in the recent work cited earlier. Although the setting of behavior cloning might be new, the paper neither proposes a new attack framework that is specially designed for behavior cloning nor provides any guarantee of its optimality, unlike previous attack papers cited above.
> > >
> > > Similarly, the authors simply applied randomized smoothing (RS) to each state as in supervised learning settings without considering any uniqueness of behavior cloning. Note that RS has been applied to policies in recent work, such as CROP: Certifying Robust Policies for Reinforcement Learning through Functional Smoothing, ICRL 2022.

---

### Official Review · Reviewer_qtnJ · 2024-11-05

**Soundness:** 2
**Presentation:** 1
**Contribution:** 2
**Rating:** 5
**Confidence:** 4

**Summary:**

This paper investigates the vulnerability of modern behavior cloning policies against popular adversarial attack methods. The behavior cloning methods includes original BC, LSTM-GMM, Implicit BC, Diffusion Policy and VQ behavior transformer. Adversarial attack methods includes projected gradient descent (PGD) and universal adversarial perturbation (UAP). Authors adapted these attack methods for some of the BC methods. The investigation is conducted on four manipulation tasks: Lift, Can, Square, and Push-T. The work explored a few interesting questions in addition: is perturbation learned transferable to policies learned using different BC methods? What is the impact of feature extraction backbone on adversarial attack and is the perturbation transferrable across different backbones. How does the action prediction horizon of diffusion policy affect its vulnerability?

**Strengths:**

1. The work provide some insights on adversarial robustness of general BC methods (DP is much more robust than other methods)
2. The work provide adaptation of classic adversarial attack methods to newer BC methods like DP, IBC, and demonstrated attack success using the adapted methods.
3. The work conducted some transferrable analysis on perturbation learned, across BC methods or across visual backbones.

**Weaknesses:**

1. The writing needs improvement in general. (see questions)
2. Some arguments are not well explained or supported by the result provided. (see questions)

**Questions:**

1. line 221: the first "maximize" should be "minimize"?

2. line 221-223: this explanation is confusing to me, why probability of selecting the targeted action low would leads to no clear loss function?

3. line 222: what does clean mean here?

4. line 229: do you mean decrease? (same for the later "increase" used)

5. Algorithm 1: the initialization of S seems redundant?

6. line 342: Q2 is not answered.

7. line 352- 356: please consider provide concise task description if possible. (even not in main paper)

8. line 363: again, this will be more clear if some concise task/benchmark description is provided somewhere.

9. line 374: what does the perturbed observation looks like, could you provide a few examples?

10. Figure 2: why report task success rate rather than attack success rate? Also, I think normal success rate refers to task success rate?

11. line 411-412: could you provide some task complexity related information? How is complexity measured?

12. line 426-427: can you explain this in more detail? A lot of factors should be considered here, for example, for more complex tasks, their task success rate are likely lower, so you might want to use some relative attack success rate to account for this.

13. line 426-427: Table 1 is success rate?, Table 2 is Mean IoU? how do you compare this two and reach the observation that "we noticed an increased propensity for adversarial perturbations to transfer between algorithms"?

14. Table 1: more explanation on what the columns and rows are? are attacks obtained in column methods and applied on row methods? also, what is "random", this is not explained in the paper I think?

15: line 460: "we observed high transferability for some algorithms (e.g., Diffusion Policy-C and VQ-BET)," are you referring to LSTM-GMM?

16: line 752- 754: The hyperparameter setting is not clear to me. First, please check the sentence. Second, what is perturbation if epsilon is set to 0.625 (which is pretty big compared to usual adversarial attack work.)

---

> ### Author Response · Authors · 2024-11-24
> **Response to reviewer qtnJ**
>
> We thank the reviewers for their helpful comments and are pleased to know that the reviewer liked our selection of algorithms and transferability studies. Comments and questions are addressed below,
> > the first "maximize" should be "minimize"?
>
> Yes. This has been fixed. (Line 234)
>
> > this explanation is confusing to me, why probability of selecting the targeted action low would leads to no clear loss function?
>
> We agree that the original wording here was confusing. We have clarified this and highlighted that the *end-to-end* loss function cannot be clearly defined, as the IBC uses iterative sampling which requires sampling the actions for a fixed amount of iterations before choosing the final action, akin to Particle Filter.
>
> > what does clean mean here?
>
> We refer to clean as the original or unattacked state-action distribution.
>
> > do you mean decrease? (same for the later "increase" used)
>
> Yes. Good catch. We have fixed this.
>
> > Algorithm 1: the initialization of S seems redundant?
>
> We have removed the redundancy.
>
> > Q2 is not answered.
>
> Our results show that attacks are easy to construct since we can get the robot to have low success across different tasks and algorithms. We do discuss the ease of implementation of attacks. In particular, we show that the explicit policies can be attacked just like any other standard supervised learning model. But that the implicit policies are more nuanced implementation wise, but still end up not being difficult, as shown by the success of our attacks in dramatically dropping the robot task completion rates. We agree this can be spelled out more explicitly. We are also running a set of ablation experiments on the attack epsilon to more clearly show the ease of attacks.
>
> > please consider provide concise task description if possible
>
> Thank you for this suggestion. We have added a brief description of the tasks in the Appendix. B.
>
> > what does the perturbed observation looks like, could you provide a few examples?
>
> We have added the examples for the rollouts with Untargeted and Targeted Universal Perturbations in the Appendix H.
>
> > why report task success rate rather than attack success rate? Also, I think normal success rate refers to task success rate?
>
> Most robotics papers report task success rate so we stuck with that convention to showcase the dramatic drop in task success when the adversarial attacks are applied. In these cases the attack success rate is simply 1-task success rate. We would be happy to plot the attack success rate in the camera-ready paper if that would be easier to interpret.
>
> > could you provide some task complexity related information? How is complexity measured?
> We have clarified at the end of section 4.3 that our task difficulty ranking is taken from prior work by Mandelkar et al. 2021 [1].
>
> > can you explain this in more detail? A lot of factors should be considered here, for example, for more complex tasks, their task success rate are likely lower, so you might want to use some relative attack success rate to account for this.
>
> Thank you very much for highlighting this. We have clarified it in Line 427-431, where we highlight the relational performance drop between Lift (less complex) and Square (more complex), while also highlighting (Line 467-470) the need for future in developing metrics that takes into account the task complexity and baseline performance.
>
> > Table 1 is success rate?, Table 2 is Mean IoU? how do you compare this two and reach the observation that "we noticed an increased propensity for adversarial perturbations to transfer between algorithms"?
>
> We have clarified this. (Line 427-431)
>
> > more explanation on what the columns and rows are? are attacks obtained in column methods and applied on row methods? also, what is "random", this is not explained in the paper I think?
>
> We have clarified the table in the caption and have also included the label for the rows and columns in the paper.
>
> > "we observed high transferability for some algorithms (e.g., Diffusion Policy-C and VQ-BET)," are you referring to LSTM-GMM?
>
> Thank you very much for catching this, the algorithms had been flipped it should have been the opposite way. We have corrected our mistake (line 478-479).
>
> > The hyperparameter setting is not clear to me. First, please check the sentence. Second, what is perturbation if epsilon is set to 0.625 (which is pretty big compared to usual adversarial attack work.)
>
> Thank you very much for catching this, there was a typo in our paper. The epsilon used is 0.0625 rather than 0.625. We have also updated the hyperparameters section (D) to be more clear.
>
> [1] Ajay Mandlekar, Danfei Xu, J. Wong, Soroush Nasiriany, Chen Wang, Rohun Kulkarni, Li Fei-Fei, Silvio Savarese, Yuke Zhu, and Roberto Mart’in-Mart’in. What matters in learning from offline human demonstrations for robot manipulation. In Conference on Robot Learning, 2021

---

> > ### Comment · Reviewer_qtnJ · 2024-11-26
> >
> > I appreciate authors' effort to address my questions, I have raised my score to 5.

---

### Official Review · Reviewer_zFjG · 2024-11-05

**Soundness:** 2
**Presentation:** 3
**Contribution:** 2
**Rating:** 5
**Confidence:** 4

**Summary:**

The paper analyses the vulnerability of imitation learning policies to adversarial attacks in the visual input space. The paper also introduces a novel adversarial attack for diffusion-based policies, as it finds that these are generally harder to attack with sota attacks.
The paper evaluates on the Robomimic environments and finds that attacks dont generalise even when similar vision backbones are used, and that diffusion policies are generally more robust.

**Strengths:**

- relevant topic and well grounded selection of attacks and imitation learning algorithm
- solid evaluation in the given environment
- novel attack for hardest-to-attack diffusion policies

**Weaknesses:**

- motivation for diffusion policies attack is unclear: when is the setting of having access to the denoising process of the policy realistic?
- the empirical evaluation is limited to just one environment (Robomicic). To draw conclusions, at least 1-2 additional environments should be considered.
- A contrasting and comparing to prior works is largely missing
- e.g. the finding that denoising results in high adversarial robustness has been made before (see "(Certified!!) Adversarial Robustness for Free!")

**Questions:**

- Could the authors specify the threat model more clearly and motivate the setting for having access to diffusion steps

---

> ### Author Response · Authors · 2024-11-24
> **Response to Reviewer zFjG**
>
> Thank you very much for your helpful comments! We appreciate that you found our evaluation to be solid and selection of algorithms well rounded.
> > motivation for diffusion policies attack is unclear: when is the setting of having access to the denoising process of the policy realistic?
>
> We focus on white-box, post-deployment attacks. We have added text to clarify this in our introduction (paragraph 5) and Section 3.1. Unlike training time attacks that corrupt the learning process, we focus on adversarial attacks on open-sourced, pretrained models. Given the increasing trend toward open-sourced ML models, we believe the types of attacks we study are a very practical concern that has not received attention in prior work. We note that we do not require access to the inner workings of the denoising process at attack deployment time. We only require knowledge of the diffusion algorithm and model parameters (white-box attack assumption). Our results show that we can significantly save on attack computation time when crafting the attack by not backpropagating into the input until the diffusion process is close to convergence.
>
> > the empirical evaluation is limited to just one environment (Robomicic). To draw conclusions, at least 1-2 additional environments should be considered.
>
> We agree that testing across multiple environments is important. We would like to emphasize that we already test our approaches across 4 different environments: 3 different environments from the Robomimic suite of environments and Push-T. Rather than just being a single environment Robomimic is designed with diversity in mind. We also already test on Push-T which is a separate and distinct domain from Robomimic. To better emphasize the variety in these tasks we have added a task description to Appendix B. We believe this is sufficient variety to demonstrate the vulnerability of the 5 different BC algorithms we consider.
>
> > A contrasting and comparing to prior works is largely missing
>
> Thank you for the reference! We have added it as a reference in the last paragraph of the introduction. We want to clarify that while there has been work on adversarial robustness of diffusion models this prior work focused on models where the input which is denoised is an image, whereas in diffusion policy the input which is denoised are actions with images conditioned on the denoising process. Additionally we have added a few more references related to our work (highlighted in blue in the intro), but we would also like to point out that the increased interest in behavior cloning policies has been recent in the robotics community and there is a lack of prior work examining the vulnerability and robustness of these methods.
>
> > Could the authors specify the threat model more clearly and motivate the setting for having access to diffusion steps
>
> We have added our threat model and clarified our setting in Section 3.1.

---

> ### Comment · Area_Chair_TZUD · 2024-11-27
> **Rebuttal Response**
>
> Dear Reviewer,
> Do you mind letting the authors know if their rebuttal has addressed your concerns and questions? Thanks!
> -AC

---

> ### Author Response · Authors · 2024-11-27
> **Additional Response to Reviewer zFjG**
>
> > The empirical evaluation is limited to just one environment (Robomicic). To draw conclusions, at least 1-2 additional environments should be considered.
>
> - As recommended, we have included an additional environment : Tool Hang. It is the toughest task in Robomimic suite of environments, as it requires precise, and dexterous, rotation-heavy movements. For more details on task description, please refer to Appendix B.5. We would like to re-iterate that we now  test our approaches across 5 different environments: 4 different environments from the Robomimic suite of environments and Push-T.
>
> - We compare and contrast the unattacked policies with the targeted universal perturbation attacked policies for Diffusion Policy, LSTM-GMM , IBC and Vanilla BC in appendix I for Tool Hang . Since VQBET policies take very long to train and then train universal perturbation attacks on them, we request that  our opennes and commitment to include new environments and the time constraints posed at the rebuttal phase are all taken into consideration.
>
> - We promise that we will have the targeted universal perturbation attacks as well as the VQBET policies themselves trained for all 3 seeds in camera ready version. We assure you that we will also included untargeted universal perturbation attacks and PGD attacks for all algorithms for 3 seeds each in camera ready version.

---

> ### Comment · Reviewer_zFjG · 2024-11-27
>
> Dear authors, thank you for the detailed response. While I remain on the fence whether the considered attack scenario on diffusion policies has real world relevance, I do agree that it is an interesting scenario worth exploring. I remain unconvinced that tasks from a single Robotic environment simulator are sufficient for a paper that -- according to its title -- studies "adversarial attacks on modern behavior cloning policies" (no further specifications). I rather see this paper as a study of attacks on imitation learning policies in Robomimic. As such, I don't think its contribution is significant. Adding environments from different environments would change this. Hence, I will maintain my score.

---

> > ### Author Response · Authors · 2024-12-01
> > **Reply to Reviewer zFjG**
> >
> > Dear Reviewer, we want to emphasize that we comprehensively test across 5 distinct environments (Lift, Can, Square, Push-T, Tool-Hang) that cover fundamentally different manipulation challenges - from basic pick-and-place to complex dexterous manipulation. Similar to how seminal RL papers often focus on either Mujoco or Atari, we use Robomimic as it is the de-facto benchmark suite for modern BC methods. Would you consider raising your score given this task diversity?

---

> ### Author Response · Authors · 2024-11-28
> **Reply to Reviewer zFjG**
>
> We appreciate your response and see your point.
> -  However, much work in RL only uses Mujoco environments or only uses Atari environments and yet is seen as general RL research. Similar to Mujoco and Atari, Robomimic contains a wide variety of environments and we believe results across these different environments provides a strong contribution in studying the vulnerabilities of modern BC policies.
>
> - While more environments are always nice, we believe environments should not just be added for the sake of having more environments. Instead environments should only be added if they will demonstrate something new. We do not think adding a new environment would significantly change our paper and would likely just reinforce the same trends we already see. If there is a specific hypothesis you have in mind that could only be tested with a different environment, please let us know.
>
> - Finally, we want to emphasize that Robomimic is the defacto test suite for modern BC methods. The only prior work closely related to our work, Diffusion Policy Attacker[1] , accepted at NeurIPS 2024, also tests the Diffusion Policy on Robomimic environments and Push-T but in contrast to our paper,  they neither investigate the vulnerability of different BC algorithms nor study the transferability of attacks across algorithms, architectures, and tasks.
>
> Thus, we believe it is the most appropriate and most compelling set of environments we can choose and, we kindly ask the reviewer to reevaluate their score in light of the additional clarifications.
>
> [1] "Diffusion Policy Attacker: Crafting Adversarial Attacks for Diffusion-based Policies", Yipu Chen, Haotian Xue, Yongxin Chen, 38th Conference on Neural Information Processing Systems (NeurIPS 2024).

---

### Author Response · Authors · 2024-11-28
**General Response to the Reviewers**

We are very thankful to the reviewers for their thoughtful comments and constructive feedback. We particularly appreciate that multiple reviewers highlighted the strength of our evaluation methodology and transferability analysis. Reviewers zFjG and qtnJ noted the solid empirical evaluation and thorough analysis of perturbation transferability across different BC methods and visual backbones. We are also grateful that reviewers zFjG and 9PBk found the paper well-organized and easy to follow.

We have made substantial improvements to address the reviewers' concerns:

1. Threat Model and Motivation:
- We have clearly defined our threat model, focusing on white-box, post-deployment attacks rather than training-time attacks
- We better motivate the practical relevance of this threat model, particularly for open-source models where parameters are publicly available
- We emphasize that unlike training-time attacks that corrupt the learning process, we study adversarial attacks on pretrained models, which is increasingly important given the trend toward open-sourced ML models

2. Expanded Evaluation:
- We have added the Tool-Hang environment, one of the most challenging tasks in the Robomimic suite, requiring precise and dexterous rotation-heavy movements
- We now comprehensively evaluate across 5 distinct environments: Lift, Can, Square, Push-T, and Tool-Hang
- We've added detailed task descriptions and complexity analysis in Appendix B

3. Additional Analysis and Defense:
- We've conducted extensive hyperparameter ablation studies for attack epsilon values (0, 4/256, 8/256, 16/256) in Appendix J
- We've implemented and evaluated randomized smoothing as a defense mechanism (Appendix F)
- We've added cross-task transferability analysis in Appendix G, showing how attacks developed for one task transfer to others
- We've included perturbed observation visualizations in Appendix H

4. Novel Findings and Contributions:
- We demonstrate that modern BC algorithms, particularly VQ-BET, are surprisingly vulnerable to small perturbations
- We show that implicit policies (IBC and Diffusion Policy) exhibit greater robustness than explicit policies
- We reveal interesting transferability patterns across algorithms and tasks
- We discover that Diffusion Policy's robustness increases with larger action horizons
- We find that traditional computer vision defenses like randomized smoothing have limited effectiveness for multi-modal action distributions

5. Technical Clarifications:
- We've fixed technical errors in equations and notation
- We've clarified hyperparameter settings and experimental procedures
- We've improved table descriptions and added detailed captions
- We've expanded our discussion of evaluation metrics and their implications

Our results provide the first comprehensive study of adversarial vulnerability across modern behavior cloning methods, showing both worst-case vulnerability (PGD) and more pragmatic vulnerability (Universal attacks). The demonstrated transferability across algorithms, vision backbones, and tasks removes the need for white-box access, making these findings particularly relevant for real-world deployment.

We believe these additions and clarifications significantly strengthen the paper and address the reviewers' main concerns while maintaining our core contribution: providing the first thorough investigation of adversarial vulnerability in modern behavior cloning methods. This work reveals important insights about the robustness of different architectures and will stimulate future research in making these increasingly deployed systems more secure.

We appreciate the reviewers effort and great suggestions and believe our paper has been significantly improved. If you agree we would appreciate it if you would change your ratings. If you do not agree, please let us know if there is anything else we can do to address your concerns and raise your scores to an accept

---

### Meta-Review · Area_Chair_TZUD · 2024-12-19

**Metareview:**

**Summary** This paper studies the vulnerability of several different behavioral cloning models  (including original BC, LSTM-GMM, IBC, Diffusion Policy (DP) and VQ-BeT) to adversarial white-box, post-deployment attacks such as PGD and UVA .  The authors show introduce a new attack for DP and IBC since standard attacks do not work as well.  They also examine the transferability of attacks between models and tasks.

**Strengths** Reviewers considered the paper to address a relevant topic and the focus on adversarial attacks in BC to be new (although attacks in policy learning have been studied before).  The paper provides insight into the varying weakness of BC methods to adversarial attacks using a variety of methods and attacks. The paper demonstrates a new attack for diffusion and implicit models which is a variant of PGD.  Lastly, the authors provide interesting empirical analysis on the transferability of attacks across BC methods, visual encoders.

**Weaknesses** One limitation of the current work is that the setting is limited to considering test-time attacks assuming knowledge of the model and weights, which may or may not be true in real world settings. Reviewer DYsQ pointed out the initial paper did not consider the effectiveness of attacks assuming methods employing modern defenses.  The authors did add random smoothing to increase robustness. Another concern was that the primary novelty of the paper is the setting of behavioral cloning with the methods being adaptations of standard methods.  Lastly, the evaluation is limited to 4 domains from robomimic and push-T.

**Conclusion** While several reviewers increased their scores, at the end they were unanimous in finding the paper borderline reject.  The paper is promising in that it provides an interesting and insightful study of a new domain, adversarial learning in BC. The authors should consider the reviewer feedback for future drafts to strengthen the paper including, adding more modern defenses, exploring other environments, considering other metrics, or developing new attacks particular to the BC setting.

**Additional Comments On Reviewer Discussion:**

zFjG engaged with the authors who added relation to prior work in revision, but did not increase their score due to lack of diversity of environments for testing. qtnj has many suggestions to improve the writing of the paper.  The authors made the changes and qtnj increased their score to 5. DYsQ asked the authors to define a threat model clearly and to consider adversarial defenses.  The authors added randomized smoothing and a definition of the threat model and DYsQ raised their score to 5.

---

### Decision · Program_Chairs · 2025-01-22

Reject